# A general framework for characterizing optimal communication in brain networks

Kayson Fakhar[1,2]*, Fatemeh Hadaeghi[2], Caio Seguin[3], Shrey Dixit[2,4,5], Arnaud Messé[2], Gorka Zamora-López[6,7], Bratislav Misic[8], Claus C Hilgetag[2,9]

[1]MRC Cognition and Brain Sciences Unit, University of Cambridge, Cambridge, United Kingdom; [2]Institute of Computational Neuroscience, University Medical Center Eppendorf-Hamburg, Hamburg University, Hamburg Center of Neuroscience, Hamburg, Germany; [3]Department of Psychological and Brain Sciences, Indiana University, Bloomington, United States; [4]Department of Psychology, Max Planck Institute for Human Cognitive and Brain Sciences, Leipzig, Germany; [5]International Max Planck Research School on Cognitive Neuroimaging, Barcelona, Spain; [6]Center for Brain and Cognition, Pompeu Fabra University, Barcelona, Spain; [7]Department of Information and Communication Technologies, Pompeu Fabra University, Barcelona, Spain; [8]McConnell Brain Imaging Centre, Montréal Neurological Institute, McGill University, Montréal, Canada; [9]Department of Health Sciences, Boston University, Boston, United States

*For correspondence:
kayson.fakhar@mrc-cbu.cam.ac.uk

Competing interest: The authors declare that no competing interests exist.

## eLife Assessment

The authors provide a **compelling** method for characterizing communication within brain networks. The study engages **important**, biologically pertinent, concerns related to the balance of dynamics and structure in assessing the focal points of brain communication. It will be of interest to researchers trying to dissect structure of complex interaction networks across scales, from cells to regions.

**Abstract** Efficient communication in brain networks is foundational for cognitive function and behavior. However, how communication efficiency is defined depends on the assumed model of signaling dynamics, e.g., shortest path signaling, random walker navigation, broadcasting, and diffusive processes. Thus, a general and model-agnostic framework for characterizing optimal neural communication is needed. We address this challenge by assigning communication efficiency through a virtual multi-site lesioning regime combined with game theory, applied to large-scale models of human brain dynamics. Our framework quantifies the exact influence each node exerts over every other, generating optimal influence maps given the underlying model of neural dynamics. These descriptions reveal how communication patterns unfold if regions are set to maximize their influence over one another. Comparing these maps with a variety of brain communication models showed that optimal communication closely resembles a broadcasting regime in which regions leverage multiple parallel channels for information dissemination. Moreover, we found that the brain's most influential regions are its rich-club, exploiting their topological vantage point by broadcasting across numerous pathways that enhance their reach even if the underlying connections are weak. Altogether, our work provides a rigorous and versatile framework for characterizing optimal brain communication, and uncovers the most influential brain regions, and the topological features underlying their influence.

## Introduction

In the realm of network science, the human brain represents a perfect example of a complex system. Viewing the brain as a network has uncovered key topological characteristics, including the presence of densely interconnected modules, central hubs, and hierarchical structures in information processing (*Bullmore and Sporns, 2009*; *Sporns et al., 2004*). This perspective, particularly obtained by analysis of large-scale structural brain networks — the connectome — elucidated that a range of network characteristics arise from the interplay between two conflicting evolutionary imperatives: minimizing wiring costs on one hand and maximizing signaling efficiency on the other (*Betzel et al., 2016a*; *Gulyás et al., 2015*; *Kaiser and Hilgetag, 2006*).

The brain is confined within a three-dimensional space, making the formation and maintenance of long-range connections metabolically expensive. Conversely, a network predominantly comprising short-range connections, although economical in wiring costs, necessitates multiple intermediate nodes for information relay to distant nodes, diminishing effective communication capacity (*Bullmore and Sporns, 2012*; *Wang and Clandinin, 2016*). It has been shown that this dichotomy leads to a compromise, where predominantly short-range connections among neighboring regions coexist with a selective set of long-range connections that function as communication shortcuts (*Bassett and Bullmore, 2017*; *Chen et al., 2013*).

In this view, signaling efficiency between nodes is graded based on the number of steps from a source to a target, presupposing information transmission *exclusively* along one path, the shortest path (*Latora and Marchiori, 2001*). However, it has been argued that such a conceptualization of signaling unrealistically requires brain regions to possess global knowledge of the network's connectivity (*Avena-Koenigsberger et al., 2017*). To address this issue, alternative **communication models (CMs)** have been developed, accommodating other conceptualizations, such as cascading dynamics and diffusive processes, where the information permeates through the network via parallel pathways. Consequently, alternative conceptualizations of how distant brain regions interact yield diverse interpretations of communication efficiency (*Seguin et al., 2023*). For instance, as mentioned above, the **shortest path efficiency (SPE)**, derived from the inverse of the shortest path length, reflects efficiency under the assumption that information flows along the shortest path only. In contrast, **diffusion efficiency (DE)**, defined by the inverse of the average path length of an unbiased random walker traversing from the source to the target, assumes information to propagate along parallel pathways with no specific preference for any of them. Quantifying how much a source can effectively influence its target depends on which perspective is chosen. Given SPE, the source has optimal influence over its unconnected target, if the number of intermediate nodes along the shortest path between them is minimal. Given DE, the source has optimal influence over its unconnected target, if it has the largest number of pathways connected with it, regardless of how long they are. Thus, the optimality of signaling depends on assumptions about the regime in which the information propagates in the network.

Moreover, despite this plurality of signaling conceptualizations, these models of propagation simplify communication patterns to linear, discrete interactions, thereby abstracting the complexities inherent in biological systems, such as non-linear interactions, oscillatory behaviors, and conductance delays (*Seguin et al., 2023*; *Suárez et al., 2020*). A plethora of biophysical models have been developed to address such details by modeling the behavior of regions as the average activity of biophysically grounded neuronal populations (*Bettinardi et al., 2017*; *Breakspear, 2017*). This mean-field formalization of brain activity comes with its own definitions of communication, such as communication through coherence, positing neuronal synchronization as a pivotal mechanism (*Buehlmann and Deco, 2010*; *Fries, 2015*; *Vinck et al., 2023*).

Therefore, while it is acknowledged that the brain's structural evolution favors optimal interregional communication, a consensus on the precise model of communication within brain networks and subsequently its optimality, remains missing due to conflicting assumptions about signal propagation. Addressing this issue is of particular interest, for several reasons. For example, the identification of structural features that improve information flow in brain networks elucidates their role in disease and may allow targeting them for specific therapeutic interventions. Such insights may also guide the design of neuromorphic systems that achieve a similar level of resource management as the brain (*Lennie, 2003*; *Padamsey and Rochefort, 2023*; *Senn et al., 2023*).

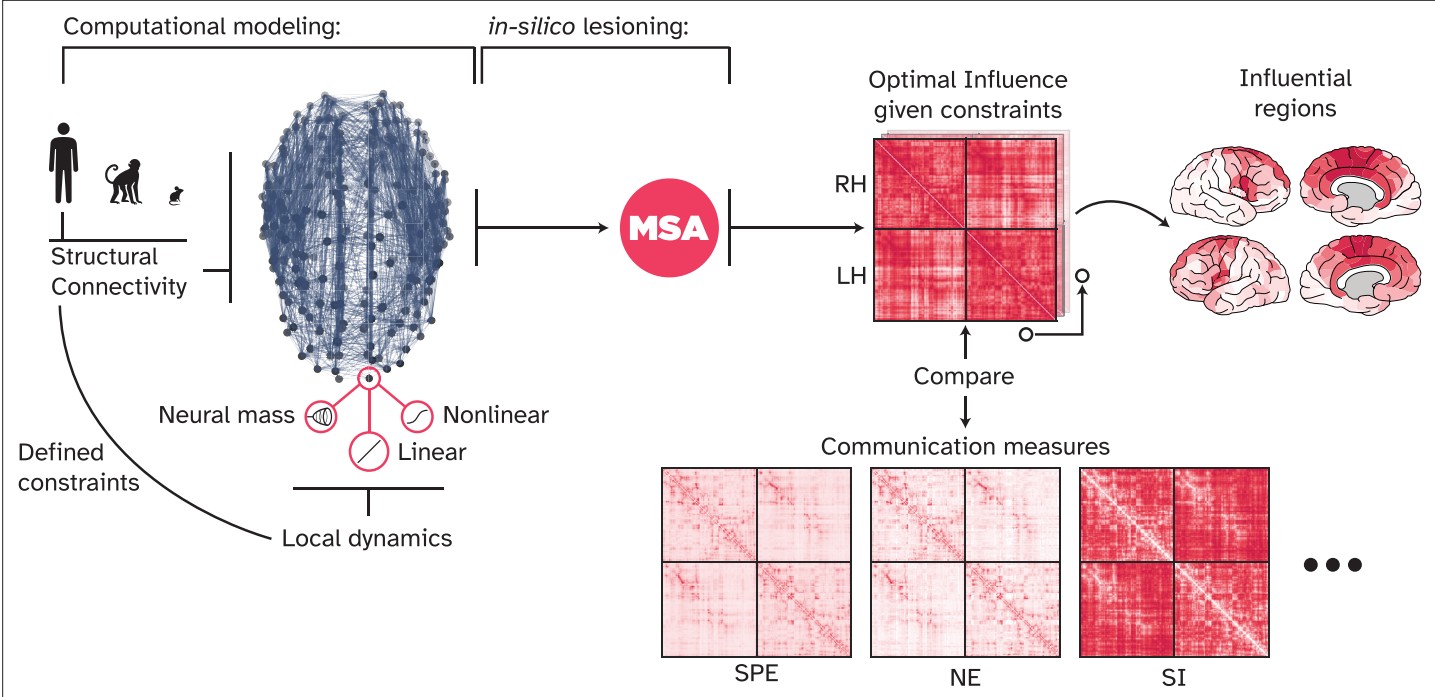

**Figure 1.** Overview of the present work. We used the human connectome and a linear model of local node dynamics as the main network model. However, other computational models can be used, building upon different assumptions of local dynamics and other network architectures. Here, we explored Macaque and mouse connectomes, as well as nonlinear models including a neural mass model of local dynamics. These two families of variables, network structure and local dynamics, define our 'game constraints,' since they dictate how information flows in the network and how nodes respond to incoming signals. For every game, i.e., a computational model such as a linear model of dynamics on the human connectome, the multi-perturbation Shapley value analysis (MSA) approach (*Figure 2*) uncovered the 'optimal influence' landscape of the network. This landscape describes a game-theoretical equilibrium point at which nodes cannot unilaterally increase their influence on each other any further. We compared the optimal influence profile of nodes at this point with various communication models and analyzed the most influential nodes in such a signal propagation regime. SPE: Shortest path efficiency, NE: Navigation efficiency, SI: Search information.

To provide a mathematically rigorous definition of optimal communication, we resort to game theory. This field analyzes interactions among agents to offer a normative account, that is, to prescribe *how they should behave* in a defined scenario, given their goals, the rules of the game, and the consequences of others' decisions (*Binmore, 2007*). Notably, game theoretical solutions are well established in fields where players lack 'agency.' In behavioral ecology, for example, game theoretical models are used to define normative descriptions for how animals should behave in scenarios involving optimizing a goal, such as foraging or reproducing (*Davies et al., 2012*; *Dugatkin and Reeve, 2000*). Thus, this theoretical approach is well suited for investigating a pivotal question in neuroscience: How should brain regions interact, considering the limitations imposed by the structural connectome and the behavior of other regions, with the aim of optimizing their communication efficiency? This question is framed by two principal constraints and one overarching objective. The first constraint is imposed by the *brain's network structure*, delineating interactions between nodes. The second constraint is the *behavior of other nodes*, dictating their response to incoming signals. The objective, in this context, is to maximize signaling efficiency – conceptualized as the extent of a node's influence over others. Such a formulation gives rise to a concrete definition of optimality as an equilibrium point where no region can increase its influence on others any further. Notably, this game-theoretical approach offers a model-agnostic and data-driven toolkit to define communication within brain networks. By 'model agnostic' we mean that the game can be constructed using any model of information propagation and local dynamics incorporated into an arbitrary connectome structure. It then defines optimality in the sense that, given the defined game, how much each node can influence other nodes. Note that optimality in this framework depends on the defined game, meaning that optimal communication can differ given different network architectures and models of communication.

In the present work (summarized in *Figure 1*), we primarily used the human structural and functional connectomes, large-scale models of dynamics, and a game-theoretical perspective of signaling to address three questions: First, how does the communication landscape look like in the state of **optimal signal propagation (OSP)**, where nodes maximize their influence on each other given the structural and dynamical constraints? Second, which model of information propagation aligns best with the data-driven formulation of signaling? And third, which brain regions are the most influential ones and why?

To answer the first question, we used **multi-perturbation Shapley value analysis (MSA)** and approximately 7000 million combinations of virtual lesions to precisely quantify the amount of influence exerted by each brain region (node) over every other node. We call this communication landscape the '**Optimal Influence (OI)**.' To answer the second question, we systematically compared putative models of neural communication against the derived OI and found **Communicability (CO)** (*Estrada and Hatano, 2008*) to capture a substantial variation observed in OI. CO is a non-conservative cascade model that posits regions to broadcast their signals across multiple parallel pathways. However, our results indicate that CO tends to underestimate the actual influence exerted via longer pathways. Thus, we proposed complementary analytical models that better replicate the OI. Further investigations using a nonlinear model of dynamics and a **neural mass model (NMM)** allowed us to determine which of the two-game constraints more strongly shape neural signaling, the global network architecture or local dynamics. We found that all models of local dynamics yielded effectively the same landscape, suggesting that network topology is a more important constraint than specific models of local dynamics. Finally, to address the third question, we revealed that areas located on the medial surface of the brain, including the precuneus, anterior cingulate cortex, and superior prefrontal cortex, exerted the strongest influence across the brain network. Notably, these regions are components of the cortical rich-club organization, which is hypothesized to act as the backbone of large-scale brain communication. Together, our results suggest that signaling in the human brain is best captured by broadcasting-like dynamics, and to optimize their influence, hub regions propagate information over multiple pathways.

## Results

### Characterizing the state of optimal signal propagation

To elucidate the concept of OSP, we start with an intuitive example. Consider a scenario where an orchestra attempts to distribute funds raised from a performance. A proposal would be to divide the funds equally among all members, a strategy that leads to the 'egalitarian value' by which every musician is paid the same amount (*Algaba et al., 2019b*). However, considering discrepancies in the effort of individual players — ranging from ensemble players to soloists and a conductor — makes this strategy *unfair* for those who contributed more. Assuming that everyone is set to maximize their share, the egalitarian proposal is rejected, and counterproposals follow until the *optimally fair solution* is found that is proportional to the invested contribution of every player. One can see this iterative process as traversing a convex solution space with a fixed point, representing the optimal solution. Any other solution is either incorrect, e.g., the solution results in shares that add up to a different value rather than the raised fund, or suboptimal, meaning that a better solution exists which, given enough time, will be selected eventually. Note that, by definition, there cannot be two equally optimal solutions into which the payoff is divided, while it is possible for multiple players to receive the same share, as with the egalitarian strategy.

This complexity mirrors the challenge we address in our study: *accurately decomposing the activity profile of each target node by identifying the precise contribution of each source node*. This problem and the space of all possible solutions for a network of three nodes is depicted in (*Figure 2A and B*). For every target node in a network of $N$ nodes, the space of all possible solutions encompasses $N$ dimensions, with the set of potential solutions having an arbitrarily complex structure. Consequently, finding the one optimal solution in this space requires a systematic and extensive exploration of different decomposition solutions. To address this issue, Shapley provided an axiomatic framework that finds the optimal solution (called the Shapley value, see Discussion about its distinction from SHAP values) using information from all possible ways the players can form coalitions (*Shapley, 1997*). This algorithm serves as an external authority that, instead of navigating the solution space using

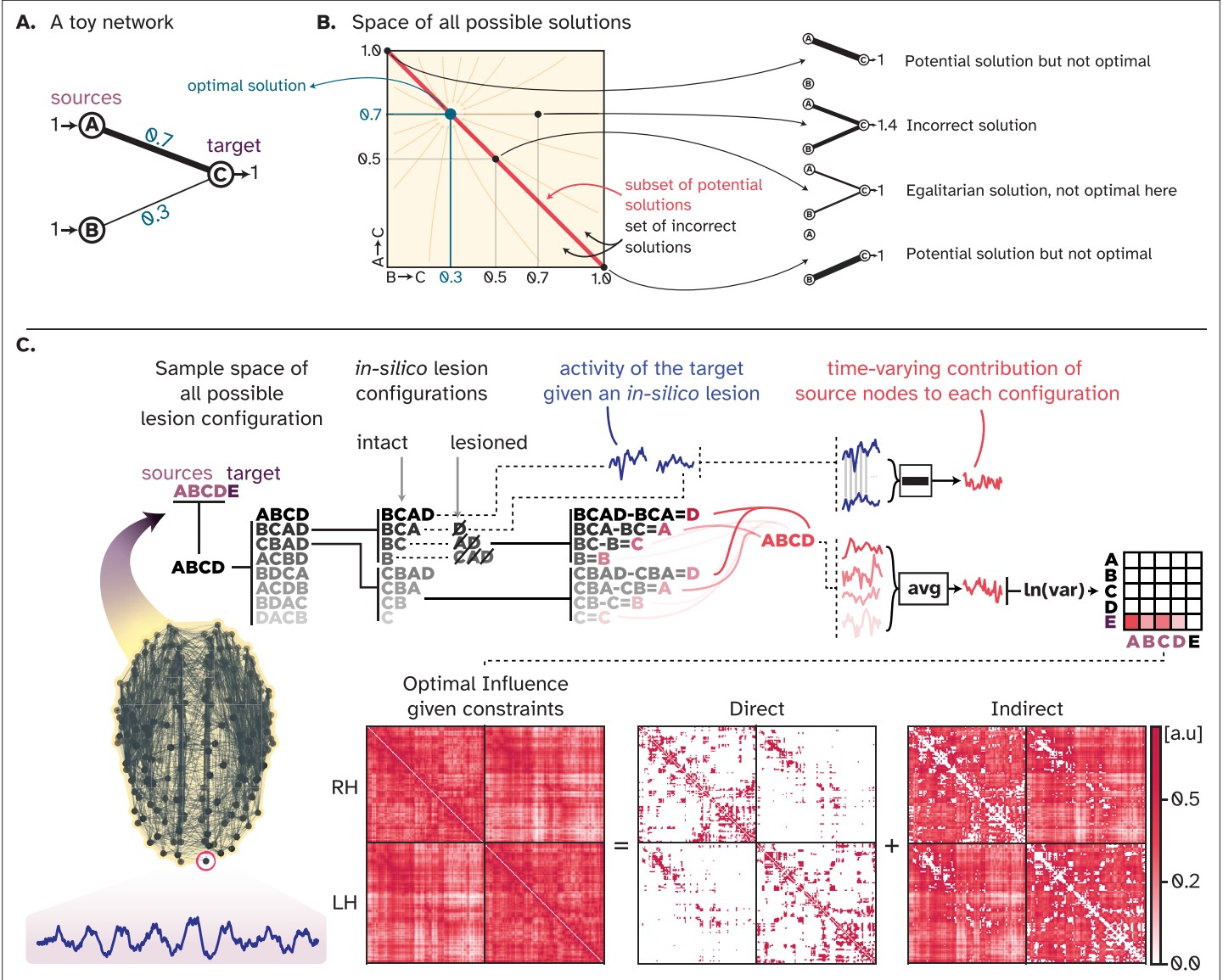

**Figure 2.** Visual summary of the multi-perturbation Shapley value analysis (MSA) approach. (**A**) A simple toy network with two sources and one target, in which the sources have different amounts of influence on the target. (**B**) For this toy network, the space of all possible ways in which the activity of the target node can be decomposed into contributions from source nodes has two dimensions. There exists a subset of potential solutions where the activity of the target node is correctly decomposed, but the decomposition is not optimal. There exists one equilibrium point where the decomposition is correct and optimal. (**C**) For every game, that is, the simulation of a whole-brain computational model, MSA performs extensive multi-site lesioning analysis to uncover the influence of every node on every other node. For every target node, MSA lesions combinations of source nodes, track how the activity of the target node changes given each perturbation and, for all source nodes, computes the difference between two scenarios: one with the source node included in the lesion-set, and the other where the source node is not lesioned. The difference between these two cases defines the contribution of the source to that specific coalition/combination of sources. Averaged over all contributions, the time-varying contribution of each source is then inferred. Iterating over all nodes results in the 'optimal influence' landscape that can be decomposed into two components: Direct influences, where nodes influence their connected neighbors, and indirect influences, where nodes influence distant unconnected nodes.

The online version of this article includes the following figure supplement(s) for figure 2:

**Figure supplement 1.** Comparing the optimal influence matrices from 1000 and 10,000 lesion samples per source node.

proposals and counterproposals, computes the contribution of each player by systematically removing them from the game and tracking the game outcome.

To do so, we used the approach of **MSA** (*Fakhar, 2021*; *Keinan et al., 2004a*) that applies exhaustive multi-site virtual lesioning across all combinations of source nodes while tracking the resultant changes in the target node, as illustrated in (*Figure 2C*) The approach is analogous to repeatedly

performing the same musical piece while each time a subset of players is excluded, thereby discerning each individual's contribution to the performed track (see, e.g. *Fakhar et al., 2024c*). For instance, the contribution of a cello player to the song is easily distinguished by taking the difference between two pieces, once with the cello player included and once without. By evaluating all possible combinations of players, higher-order interactions are then accounted for (e.g. contributions of the entire string ensemble), and thus the exact contribution of each player across all possible configurations is inferred (see section *Game-theoretical Framework* in Materials and Methods for more details). In essence, our framework reveals the unique itemized description of each player's contribution to the overall outcome. This detailed analysis allows for a precise division of payoffs, ensuring that players have no incentive to deviate from their allocated share (*Gul, 1989*; *Pérez-Castrillo and Wettstein, 2001*). The resulting game-theoretical equilibrium state, here termed the optimal signal propagation state, represents the only fair allocation of nodal influence in which no better solution can be found. Note that the 'equilibrium' here refers to the unique point in the solution space of the division problem and not the neural space (See Discussion). The MSA begins by defining a 'game.' To derive OSP, this game is formulated as a model of dynamics, such as a network of interacting nodes. These can range from abstract epidemic and excitable models (*Garcia et al., 2012*; *Messé et al., 2015a*) to detailed spiking neural networks (*Pronold et al., 2024*) and to mean-field models of the whole brain dynamics, as chosen here (see below). The model should ideally be fitted to reflect real data dynamics, after which MSA systematically lesions all nodes to derive the OSP. Put together, the framework is general and model-agnostic in the sense that it accommodates a wide range of network models built on different empirical datasets, from human neuroimaging and electrophysiology to invertebrate calcium imaging and anything in between. In essence, the framework is not bounded to specific modeling paradigms, which then allows direct comparison among different models (e.g. see section Global Network Topology is More Influential Than Local Node Dynamics).

Specifically, in this work, the 'game' was conceptualized as a large-scale model of the brain dynamics approximating the empirical **functional connectivity (FC)** by associating a differential equation to each node of the given **structural connectivity (SC)**. By utilizing the consensus SC and averaged FC data from 70 healthy young individuals (*Griffa et al., 2019*; *Shafiei et al., 2019*), we calibrated a linearized Wilson-Cowan model, as detailed in the 'Large-scale Computational Models of the Brain

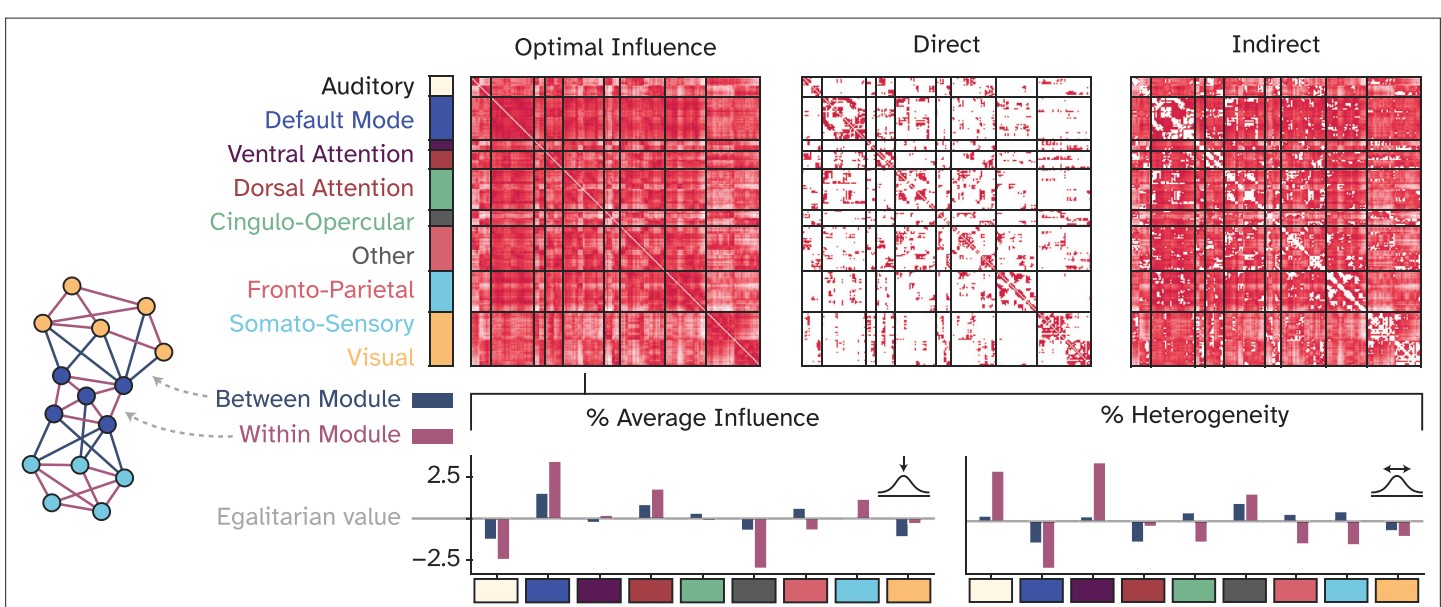

**Figure 3.** Large-scale network organization of the optimal influence landscape. Optimal influences were reorganized according to nine large-scale functional network modules and characterized by two metrics: 1. Average influence, which captures the normalized total influence of a given module within itself and to other modules. 2. Heterogeneity of influences, accounting for the normalized variation of influences within a functional module and to other modules. The egalitarian value represents a naive strategy for assigning contributions in which all players are assumed to contribute equally, i.e., by equal division of the total influence among all nine functional modules. The bars represent how much each module deviates positively or negatively from the egalitarian assumption.

Dynamics' section under Materials and methods. Then, MSA was used to perform systematic in-silico lesioning of combinations of source nodes while tracking the altered activity of a target node. This iterative process over all nodes elucidates the comprehensive influence landscape of every node on every other node within the network (illustrated in *Figure 2*). This landscape depicts the extent of influence one node exerts over another under the state of OSP.

*Figure 2* reveals that brain regions exert influence not only on their directly connected neighbors, but also on distant and unconnected regions. This observation suggests that optimal communication within the brain requires the propagation of signals beyond immediate connections, traversing through intermediate regions to form multi-step processing pathways. This notion is supported by evidence of significant functional coupling between spatially distant, unconnected regions (*Suárez et al., 2020*). We then reorganized the OI matrix according to established large-scale functional brain modules, i.e., the resting-state networks, as depicted in *Figure 3*. We calculated two key metrics: firstly, the average intra- and inter-module influence, which quantifies the normalized aggregate influence that regions belonging to a module exert both internally and on other modules. Second, we assessed the heterogeneity of these influences to determine the diversity of communication patterns within and between modules. A low heterogeneity indicates a narrow distribution of normalized total influence among nodes, measured as the standard deviation, whereas high heterogeneity suggests a wide variation in interaction strengths. We then asked *whether any modules exhibited deviations from the expected levels of influence and heterogeneity*. We derived the expected level via the egalitarian strategy where influence is uniformly distributed across all modules, corresponding to an equal division of 100% of the flow among the nine modules. Our analysis revealed a generally balanced communication landscape, with the largest deviations being approximately 3%. However, the default mode network stood out, demonstrating both substantial average intra- and inter-module influence and lower heterogeneity (*Figure 3*). This finding implies that nodes in the default mode network have a large and relatively similar amount of influence over each other and the rest of the brain. In contrast, the auditory network is characterized by a more varied and modest level of influence, predominantly within its own module, but extending to others as well. Together, the results presented in *Figure 3* suggest that while there is a relatively homogeneous level of influence within and between different cognitive modules, the default mode network distinguishes itself as a key orchestrator of brain-wide communication dynamics.

Collectively, our findings in this section indicate that manipulating a single region can impact the global dynamics of the entire brain (also as shown by *Grayson et al., 2016*; *Rabuffo et al., 2023*; *Young et al., 2000*), extending beyond anatomically connected regions. As a result, a critical question arises: *Which poly-synaptic signaling conceptualization most accurately encapsulates these observed indirect influences?*

## Optimal signaling via broadcasting over parallel pathways

To determine which model of signal propagation most accurately captures the indirect influence exerted by brain regions on one another, we explored a spectrum of communication models, spanning from routing strategies to diffusive processes (*Seguin et al., 2023*; *Seguin et al., 2020*). Routing strategies postulate that information travels predominantly along, or with respect to the shortest path. As introduced before, SPE is straightforward in its approach, focusing on the shortest path alone. **Navigation Efficiency (NE)**, on the other hand, is a geometrically greedy routing model, aiming to minimize the physical distance to a target node at each step. **Search Information (SI)** quantifies the amount of information required for a random walker to find and navigate along the shortest path. Therefore, the more the required information, the more difficult the communication between two nodes is assumed to be. Conversely, at the opposite end of the spectrum lie diffusion processes that envision information cascades throughout the network. This could involve longer, even recurrent routes to the same nodes. The efficiency of signaling under these processes is then quantified by CO and DE. CO adopts a broadcasting strategy, whereby information spreads from the source and reaches the target through multiple pathways, while being subjected to exponential decay with each step. This decay inherently reduces the impact of longer pathways. DE, in contrast, does not incorporate attenuation, focusing instead on counting the steps a random walker takes from source to target. For detailed descriptions of these models, refer to the 'Communication models' section under Materials and methods.

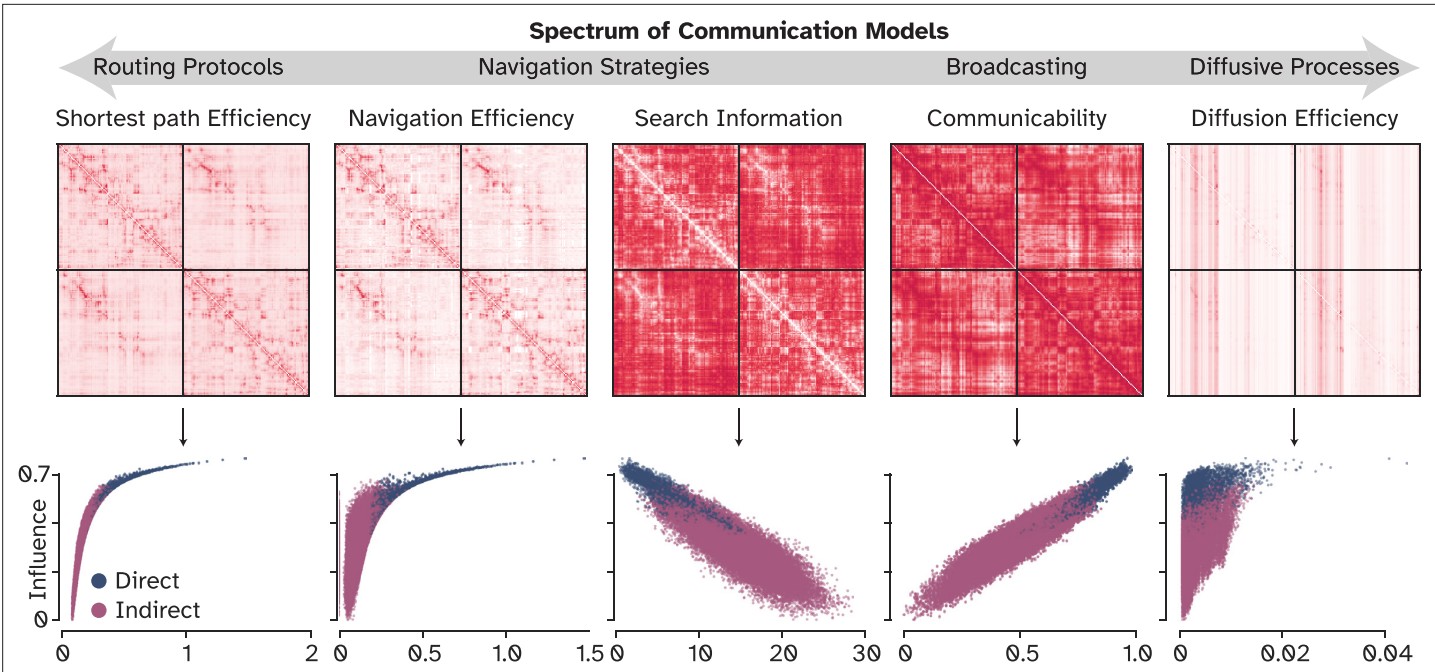

**Figure 4.** Comparison of optimal influences with other putative communication measures. Pairwise Pearson's correlation between each communication measure and optimal influences, to characterize the alignment with existing communication models. Communicability ($r$=0.94) and Search Information ($r$=−0.83) had the best alignment in this univariate setting.

The online version of this article includes the following figure supplement(s) for figure 4:

**Figure supplement 1.** Comparing optimal influence with functional connectivity.

A simple pairwise Pearson's correlation between OI and each CM matrix showed a large positive correlation between OI and CO ($R^2$=0.88; all p-values in this work are strictly smaller than 0.0001 unless mentioned otherwise) and a large negative relationship with SI ($R^2$=0.68) while other metrics such as SPE, NE, and DE exhibited nonlinear relationships with OI. This finding (*Figure 4*) suggests that, although all conceptualizations capture a degree of OI, the one assuming an attenuating diffusion dynamic for the signal propagation over multiple pathways most effectively mirrors OI. This result aligns with existing studies indicating CO's efficacy in capturing the spread of electrical stimulation to unconnected nodes (*Seguin et al., 2022a*), predicting large-scale functional modules more accurately than SC (*Seguin et al., 2022b*), and determining the degree of compensation by other regions following perturbation (*Betzel et al., 2016b*).

However, we noticed that CO slightly underestimates the true influence of longer pathways. To address this limitation, three alternative approaches were explored: adjusting communicability with a scaling factor (**scaled communicability; scaled CO**) (*Zamora-López et al., 2016*), using a linear instead of exponential attenuation (**Linear Attenuation Model; LAM**), and employing the covariance structure of a **spatial autoregressive model (SAR)** that considers linear attenuation and regional co-fluctuation. Each of these models incorporates a single adjustable parameter: a scaling factor for scaled CO, an attenuation factor for LAM, and a degree of spatial influence for SAR. After fitting these parameters to replicate OI, we found that, except for SAR, the other two models failed to rectify the issue (as shown in *Figure 5A*). While scaled CO and LAM showed marginal improvements in variance explained ($R^2$=0.89 and 0.9, respectively) compared to CO ($R^2$=0.88), SAR achieved a near-perfect performance ($R^2$=0.997). We then fitted SAR to CO instead of OI to evaluate to which extent CO captures the real degree of spatial influence. Varying the SAR parameter indicated that CO's positioning on the spectrum is considerably skewed towards a steep attenuation (0.06), in contrast to OI's more moderate rate of spatial decay (0.43). This finding supports our intuition that CO is overly strict with discounting longer pathways, since it assumes the signal to fade almost seven times faster than it actually does. It also indicates that a wide range of discount factors between 0.2 and 0.7 performs

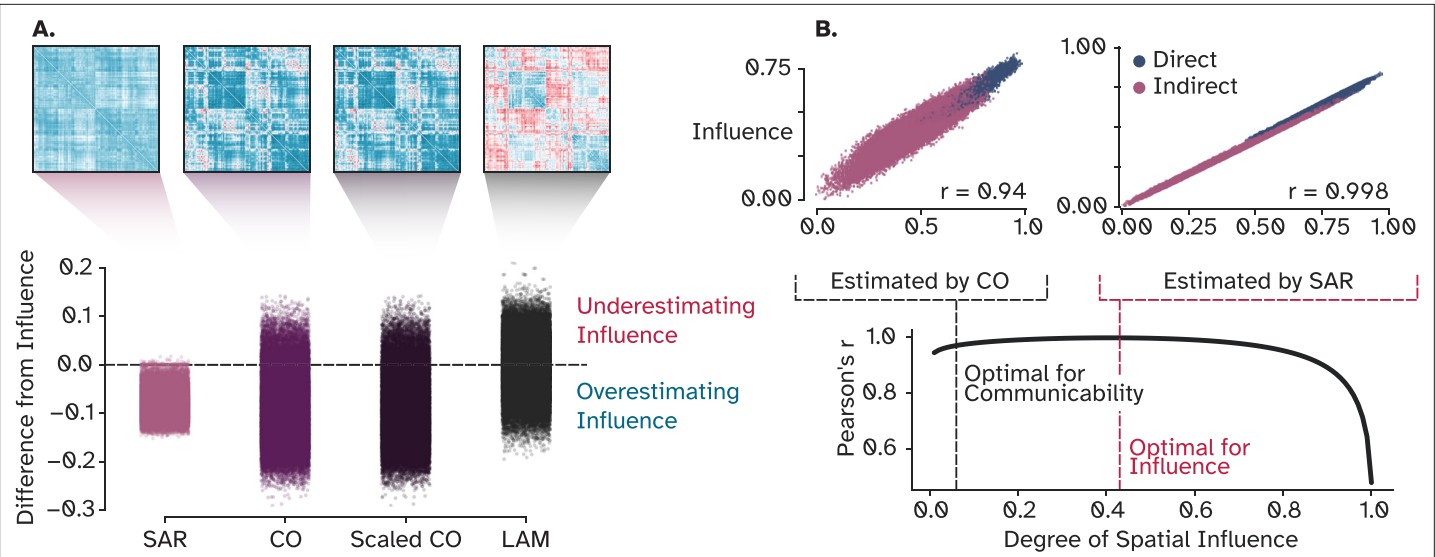

**Figure 5.** Strategies to alleviate the underestimated influence of long-range pathways. (**A**) Difference between adjacency matrices of tunable models and OI that indicates where these models underestimate (red) and overestimate (blue) the influence of nodes on each other. (**B**) SAR obtains a correlation coefficient of *r*=0.99 by increasing the depth by which signals can travel, however, the plot on the bottom shows that a wide range of values from 0.2 to 0.7 can be chosen with a small degradation of fit. SAR: Spatial Auto-Regressive model, CO: Communicability, OI: Optimal Influence.

The online version of this article includes the following figure supplement(s) for figure 5:

**Figure supplement 1.** Fitting curve of the scaled communicability (left) and the linear attenuation model (right) to the optimal influence matrix of the human connectome.

**Figure supplement 2.** Fitting curves of the spatial autoregressive model (top), scaled communicability (middle) and the linear attenuation model (bottom) to the optimal influence matrix of macaque (left) and mouse (right) connectomes.

similarly well (*Figure 5B*), relaxing the need to search for an exact value (same applies to the scaling factor of CO and attenuation factor of LAM; see *Figure 5—figure supplement 1*).

We subsequently explored the depth of influence in terms of the number of hops, specifically addressing how far a signal travels before its causal impact substantially diminishes. To this end, we used a linear regression model, fitted it to results from progressively diffused signals in the network. Put simply, we iteratively predicted the OI starting from walks of zero length (no spread of influence) and incrementally considered longer walks, applying a monotonic discount to additional steps. The analysis, as depicted in (*Figure 6A*), demonstrated peak performance at the sixth step, achieving an R² of 0.75. This means that the summation of only the first six steps captures 75% of the variance in OI (also see *Figure 6—figure supplement 1* for the result from an exponential discount instead of linear). Furthermore, a multivariate Lasso regularized linear regression model was trained on all walks simultaneously to identify a parsimonious combination that most effectively predicts OI (*Figure 6B*). Aligning with the result from the univariate model, walks of length five, six, and the eighth steps (excluding seventh) were contributing the most, allowing the model to predict OI with a high degree of accuracy (R²=0.96). These results imply that signal influence notably decreases after approximately eight processing steps, despite the network's diameter being around ten steps, meaning that the information flow between the most distant nodes is heavily degraded (*Figure 6B*). While information from the initial eight steps is sufficient to explain 96% of the OI, the superior predictive performance of models such as SAR, LAM, and CO suggests that incorporating all possible paths yields additional, albeit small, information. Moreover, the real advantage of communication models lies in their relative simplicity compared to using a regularized multivariate model. Altogether, it is important to note that the longer processing pathways that are missed by CO, Scaled CO, and LAM still influence the target node, but based on comparing how much SAR gained in predictive performance, they account for about 5% of the whole communication dynamics. This finding provides a rough estimate of how much mismatch, and in which pathways, are expected when using CO, scaled CO, and LAM in human neuroimaging data. However, when these communication models were applied to mouse and macaque connectomes, a discrepancy in capturing OI in directed networks emerged (*Figure 5—figure*

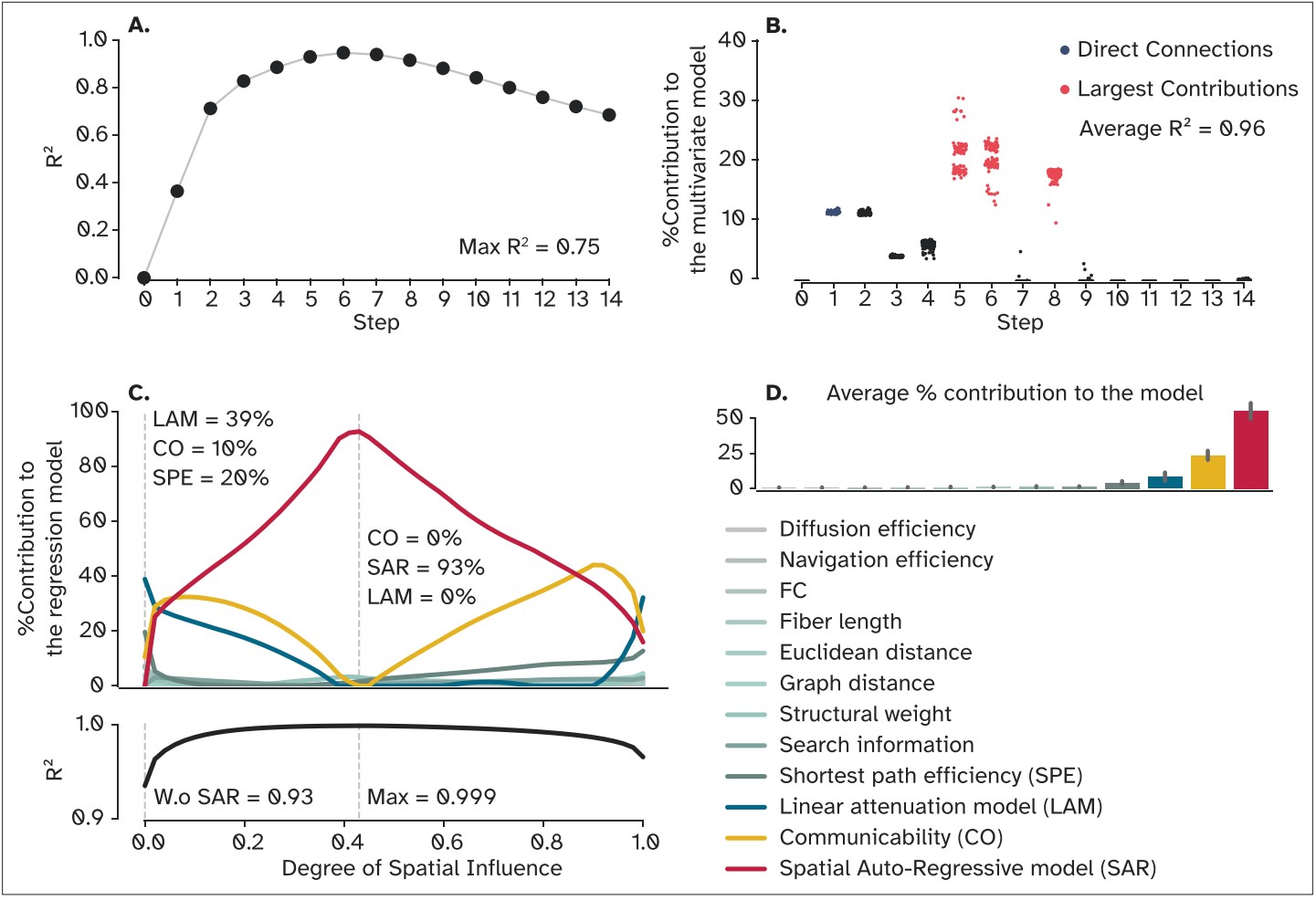

**Figure 6.** Uni-and-multivariate models of OI. (**A**) Predictive power of a linear regression model trained on walks up to 15 broadcasting steps. (**B**) A regularized multivariate model trained on all steps shows a small set of steps to be sufficient to explain 96% of OI on average. Each dot represents one cross-validation trial (N=100). (**C**) Sets of regularized multivariate models were trained on network features at each degree of spatial influence for spatial autoregressive model (SAR) and linear attenuation model (LAM) to capture the impact of this parameter on the performance of the statistical model. The dashed line on the left shows a scenario when the degree of spatial influence is zero, effectively rendering SAR an identity matrix. The second dashed line indicates where the model performed the best, explaining 99% OI. (**D**) Bars represent the contribution of each feature to the model, averaged over degrees of influence. Error bars indicate the 95% confidence interval. OI: Optimal influence.

The online version of this article includes the following figure supplement(s) for figure 6:

**Figure supplement 1.** The amount of which univariate and multivariate models can predict optimal influence from summing individual walks, discounted exponentially (as with communicability).

supplement 2). SAR marginally outperformed LAM and scaled CO in the mouse connectome ($r$=0.21, 0.18, 0.17, respectively) but showed significantly better performance in the macaque ($r$=0.92, 0.24, 0.23, respectively). The enhanced performance from mouse to macaque hints at the potential influence of network size, considering the macaque network comprises 29 nodes compared to the mouse's 112. Nonetheless, the specific effects of network size, density, reciprocity, weight distribution, and normalization, as well as other global characteristics on modeling OI necessitate further systematic investigation using synthetic network models. This finding delineates the limitations of current CMs in accurately depicting optimal signal propagation in large, directed networks. However, the observed inverted U-shaped fitting curves of LAM and scaled CO (*Figure 5—figure supplement 2*) suggest a slower spatial decay in signal propagation than assumed by models like CO. In essence, despite the struggle of these models to precisely capture the optimal signal flow in larger directed networks, there is a consensus that the influence of regions extends further than traditionally postulated by communication models.

In our final analysis of this section, we sought to determine if other communication models, despite their complex relationship with OI, could surpass CO in predicting it. Additionally, it is possible for more intuitive variables, such as the fiber length or Euclidean distance, to provide better predictive performance. To see if OI is better predicted using available variables, including CMs, another multivariate regularized linear regression model was employed (illustrated in *Figure 6C*). Excluding SAR, the model successfully explained 93% of the variance in OI using just three variables: LAM, CO, and SPE, with LAM making the most significant contribution. We then trained a set of models with the same variables but different attenuation factors of LAM and SAR to observe the dynamics of feature importance across different factors. To our surprise, at its optimal parameter value, SAR emerged as the predominant predictor, effectively simplifying the statistical model to a univariate format and once again perfectly predicting OI (as seen in *Figure 6C*). SAR maintained its dominance as the most critical variable over a range from 0.1 to 0.8, after which CO became more influential. This shift underscores the role of discounting the signal, elucidating why DE was the least effective model in our study. The reason is that, at around $\alpha = 1$ where no discount is considered, SAR converges to DE, rendering it uninformative, whereas LAM retains its distinctiveness, surpassing both CO and SAR. Averaged over all values of $\alpha$ shows that SAR was the most important feature, followed by CO and LAM (*Figure 6D*) which all follow broadcasting-like communication dynamics .

Collectively, our findings indicate that optimal influence in brain networks requires signal propagation across multiple pathways, akin to a broadcasting approach. The influence diminishes with each processing step, but not as quickly as it is assumed by CO. Moreover, a simple statistical model, i.e., SAR, accurately predicts the network's influence structure at the optimal signal propagation regime. Importantly, previous studies indicated that SAR performs as well as many other biophysically detailed NMMs in predicting the FC (*Messé et al., 2014*; *Messé et al., 2015b*). Interestingly, comparing the performance of multivariate models against simple CMs suggests that little is gained from additional variables and more computationally expensive predictive models.

It is important to acknowledge that all CMs, including SAR, are predicated on a linear dynamic model, prioritizing simplicity over biophysical complexity. However, brain regions are argued to perform nonlinear operations, show oscillatory patterns of activity and show metastability by switching between multiple states to support cognition and behavior. Therefore, to affirm the notion of optimal

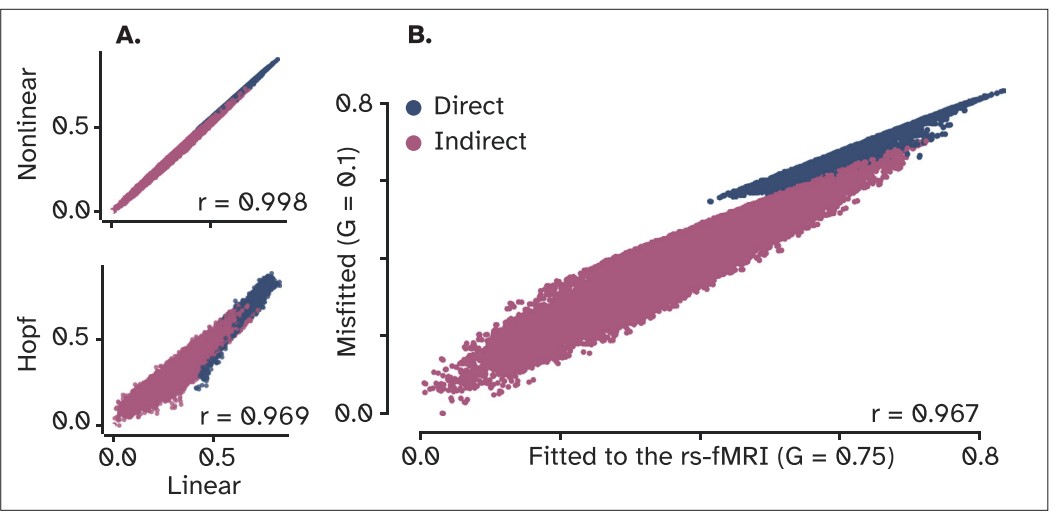

**Figure 7.** Correlations among linear, nonlinear, and neural mass models of local dynamics. (**A**) OI matrices for a linear and a nonlinear model show a near-perfect correlation. Moreover, a Hopf model also has a strong correlation with the linear model, indicating that a linear model of local dynamics represents the communication in brain networks under optimal influence (OI) as well as more complex node models (**B**) Scatter plot between two settings of the same linear model, one fitted to resting-state fMRI and the other using a random value for the global coupling parameter. The finding provides evidence for the greater role of network's structure compared to nodal dynamics in shaping brain-wide optimal influence.

The online version of this article includes the following figure supplement(s) for figure 7:

**Figure supplement 1.** Temporal characteristics of optimal influence, given the Hopf model.

communication in brain networks, it is essential to compare these findings with more realistic dynamics that capture such complex phenomena.

## Global network topology is more influential than local node dynamics

Previous research has demonstrated that linear models of local dynamics are as effective, if not more so, as nonlinear models in capturing the macroscopic dynamics of the brain (*Abdelnour et al., 2014*; *Messé et al., 2023*; *Messé et al., 2014*; *Nozari et al., 2024*). This observation led us to hypothesize that optimal signal propagation in large-scale brain networks, as measured by fMRI-BOLD signal, might also be adequately represented by linear communication models. In other words, we hypothesized that *optimal signal propagation should follow the same line of reasoning and be relatively consistent across different dynamical models*. The equilibrium state in decomposing contributions should remain consistent across different dynamical models, irrespective of their complexity. This expectation would hold true unless the local dynamics — how regions process incoming signals — impose substantial constraints on information flow within the network.

To evaluate this hypothesis, we incorporated two additional models into our analysis: one using a nonlinear **tangent hyperbolic (Tanh)** transfer function, contrasting it with the linear function used in the linear model, and the other one a well-established oscillatory NMM known as the Hopf/Stuart-Landau model, operating in its critical metastable regime (*Deco et al., 2017*). Pearson's correlation conducted to compare these models with the linear model confirmed our hypothesis. As depicted in (*Figure 7A*), both the nonlinear and Hopf models showed high correlations with the linear model ($R^2$=0.98 and 0.94, respectively). Furthermore, we investigated the influence of local dynamics on OI by contrasting a model fitted to FC with one that was not, hence producing dynamics that poorly resemble the FC. The resulting scatter plot of OI for these models, as shown in (*Figure 7B*), indicated another strong correlation ($R^2$=0.93). Collectively, these findings suggest that the equilibrium state of optimal signal propagation does not substantially vary even when employing more complex dynamical models. The reason is that, as previously found *Fakhar et al., 2024c*, while nonlinear transformations significantly impact the *time-varying* structure of individual contributions, they do not alter the overall amplitude of them, which in this context, is viewed as the *total* amount of influence one node exerts over another. Scaling, on the other hand, that determines the magnitude of these contributions, is inherently governed by the structural connectome across all models. Consequently, *the total influence* of one node on another is predominantly dictated by network topology rather than the specific dynamical model employed. However, *the time-resolved* pattern of fluctuations is specified by the model, which here is simply neglected as a direct consequence of collapsing the temporal information (refer to *Figure 7—figure supplement 1* for more details).

In summary, our findings indicate that linear models of dynamics effectively capture the degree of influence regions exert on each other under the optimal signal propagation state. Findings from this section not only corroborate previous research suggesting that macroscopic dynamics of large-scale brain networks are well approximated by linear models, but also support the hypothesis posited in the preceding section. Together, these two aspects suggest that it is the structural connectivity that shapes the landscape of optimal signaling in brain networks, rather than the models of local dynamics, since OIs are remarkably similar irrespective of the biophysical complexity embedded in models of local dynamics. The equilibrium reached in this game-theoretical framework, as demonstrated, is accurately represented by simplified abstract models of signal propagation. Thus, not only do simple communication models such as SAR perform on a par with NMMs in predicting empirical FC, but also, they represent how information optimally flows in large-scale brain networks.

## Rich-clubs harness parallel pathways to amplify their influence

It has been shown that a set of cortical hubs exerts a relatively large wiring cost to link with other distant regions, forming a rich-club organization (*van den Heuvel and Sporns, 2011*; *Zamora-López et al., 2011*; *Zamora-López et al., 2010*). Thus, these topologically central nodes are argued to be the backbone of information transfer across multiple distant specialized modules (*Harriger et al., 2012*; *van den Heuvel et al., 2012*). Previous works proposed that these connections play a crucial role in information processing by diversifying areal input-output connectivity (*Betzel and Bassett, 2018*), leading to a richer functional repertoire (*Zamora-López et al., 2016*), or integrating information from the whole brain (*Goulas et al., 2015*). Moreover, it was shown that the connections

among adjacent regions are stronger than those among distant areas (*Beul et al., 2018*; *Lynn et al., 2024*). Empirical findings and modeling work have suggested an exponential distant rule by which the strength of connections decays exponentially with spatial distance (*Ercsey-Ravasz et al., 2013*; *Markov et al., 2013*). For instance, in our human connectome dataset, the correlation between structural weights and fiber lengths is –0.6 (See *Figure 8—figure supplement 1*). Together, these findings pose a question, *if and how do rich-club regions keep their line of communication optimal given these relatively weak connection strengths of long-distance projections?*

To answer this question, we first investigated the relationship between the strength of connections between pairs of connected regions and the respective influence they assert on each other. Intuitively, the stronger a connection is, the greater its influence, leading to stronger communication between two regions. This relation has also been supported experimentally by comparing the effect

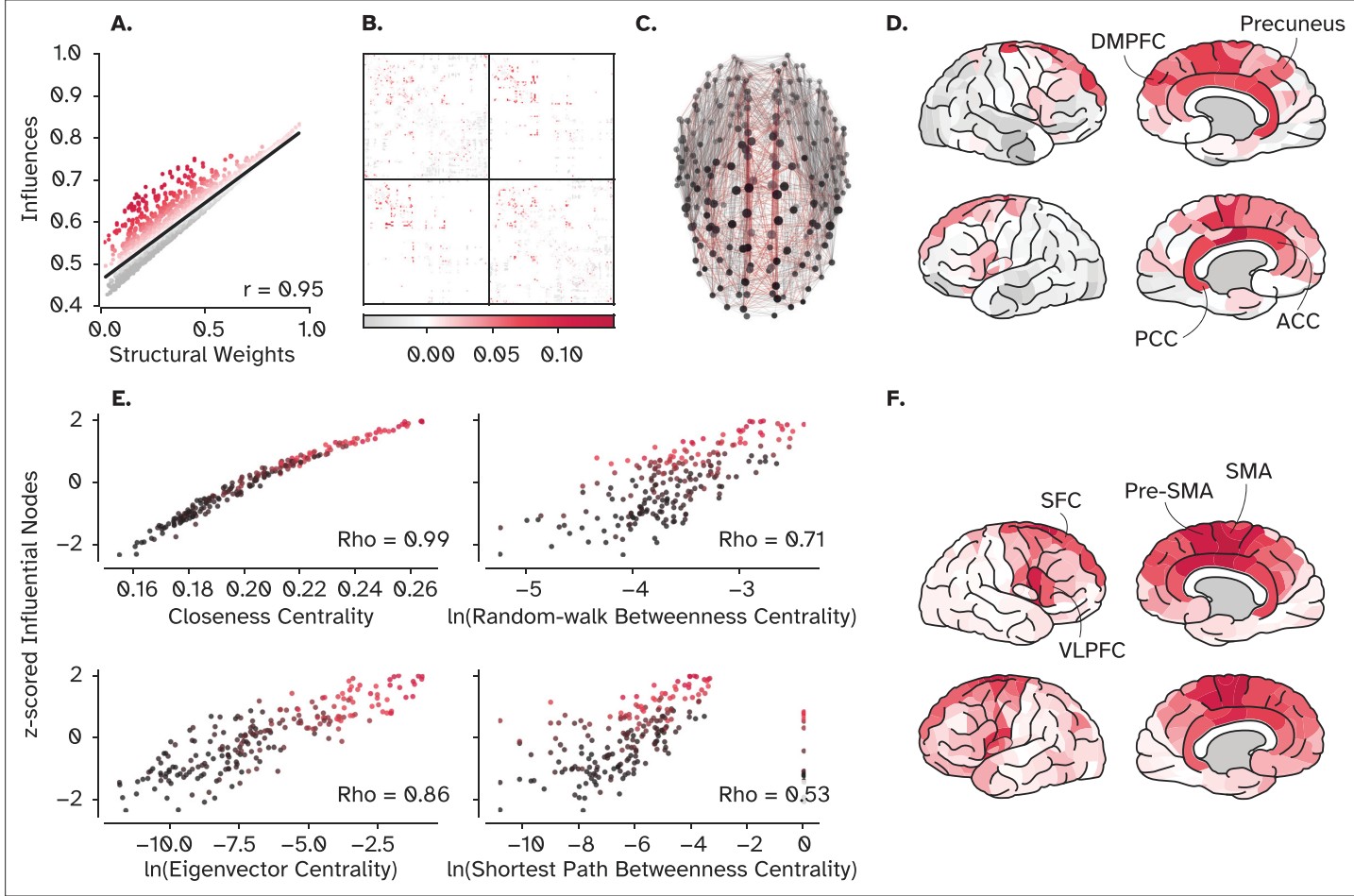

**Figure 8.** Hubs, rich-club nodes, and their influence. (**A**) Scatter plot of the influences of nodes on their connected neighbors (direct influence in *Figure 2*) versus the strength of the underlying connections. Despite the overall strong correlation between the total influence and the strength of structural connections, residuals (depicted by warm colors) reveal several weak connections with more influence than expected by their connection weights. (**B**) The position of weak connections with strong influence in the adjacency matrix. (**C**) The position of these connections in the structural brain network shows that they are mainly long-range connections among the hub regions of the cortex. (**D**) shows those hub regions. (**E**) The relationship between the influence of a region and its centrality, by four centrality measures (see Material and Methods, Graph-theoretical Measures). (**F**) The most indirectly influential nodes (indicated by red points in E) that can propagate information to distant and unconnected regions. DMPFC: Dorsomedial prefrontal cortex, PCC: Posterior cingulate cortex, ACC: Anterior cingulate cortex, SFC: Superior frontal cortex, SMA: Supplementary motor area, VLPFC: Ventrolateral prefrontal cortex.

The online version of this article includes the following figure supplement(s) for figure 8:

**Figure supplement 1.** The relationship between the log-normalized structural weights and the fiber length, which had a correlation of –0.6, suggesting that the longer the fiber, the weaker its strength is.

**Figure supplement 2.** Further analysis of the influence of weak connections in null, synthetic, and supplementary networks.

of optogenetics stimulation of a region over its connected neighbors (*Kim et al., 2023*). Our result corroborated this finding and showed a strong correlation between the structural weight and the amount of influence nodes assert on their connected neighbors (*r*=0.95). However, several pair-wise interactions fall well above the regression line (*Figure 8A*), suggesting that some regions assert strong influence on others despite their weak connection weights (likewise noticeable in both mouse and macaque connectomes; *Figure 8—figure supplement 2A*). This 'bump' is also captured by SAR, LAM, and CO (*Figure 8—figure supplement 2B*), providing further evidence that these simple models can reproduce optimal information flow among connected regions. As (*Figure 8B, C and D*) shows, these connections mostly lie between regions located at the medial surface of the cortex, coinciding with rich-club regions (*van den Heuvel et al., 2010*; *van den Heuvel and Sporns, 2011*). We then computed the amount that each node influences other *unconnected* regions (*Figure 8F*) and found the same regions to be the most influential, providing more evidence that this influence is not related to the weak link itself, but the node's overall connectivity. To test this hypothesis, we first computed several graph centrality measures, built both on the shortest-path and random walk signaling conceptualization. A Spearman's rank correlation between how influential a node is and how central it is suggests two conclusions (*Figure 8E*): First, all centrality measures show a moderate to strong positive correlation with the amount of influence nodes assert over the rest of the network, directly and indirectly. Second, those with the largest *indirect* influence (indicated with red) always lie on the top-right of the data cloud, supporting the idea that they are topologically central. We further utilized null and synthetic networks to see if we can delineate the role of connection strength and connectivity pattern in signal propagation. Although less noticeable, the trend persisted when the network topology was shuffled while nodal strength was preserved. It also persisted when the weights were shuffled while

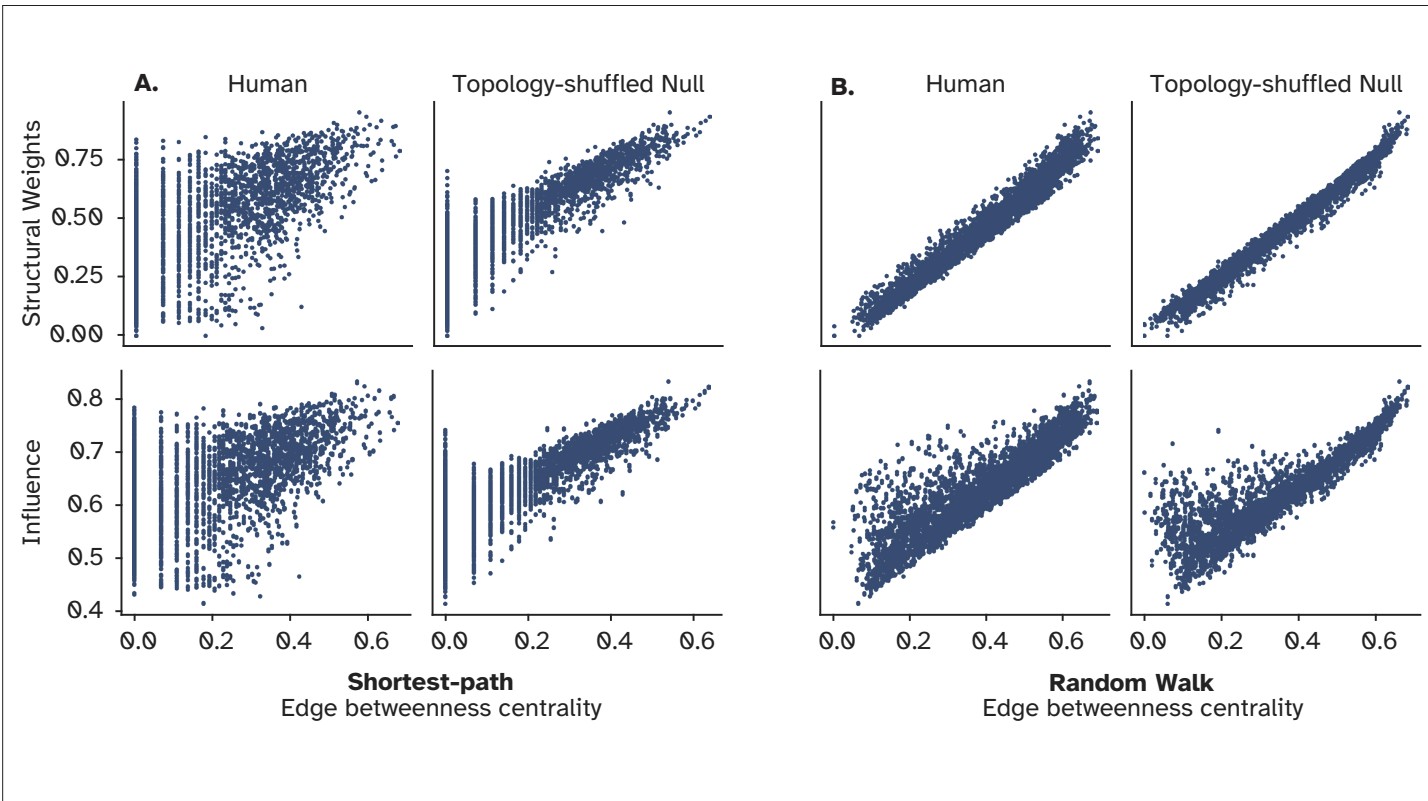

**Figure 9.** Relationship between the centrality of individual connections and their influence. (**A**) Strength of individual connections versus their centrality in terms of shortest-path edge betweenness centrality. The lower panel shows the same relationship, but for the amount of influence a node has over its connected neighbor compared to the centrality of their connections. (**B**) Depicts the same information for the random walk (diffusive) model of information flow. In both A and B, this relationship is compared against a null model in which the structural connectivity is shuffled while preserving the strength of each node, i.e., the weighted sum of its connections.

The online version of this article includes the following figure supplement(s) for figure 9:

**Figure supplement 1.** The relationship between influential nodes (in red) and their controllability measures.

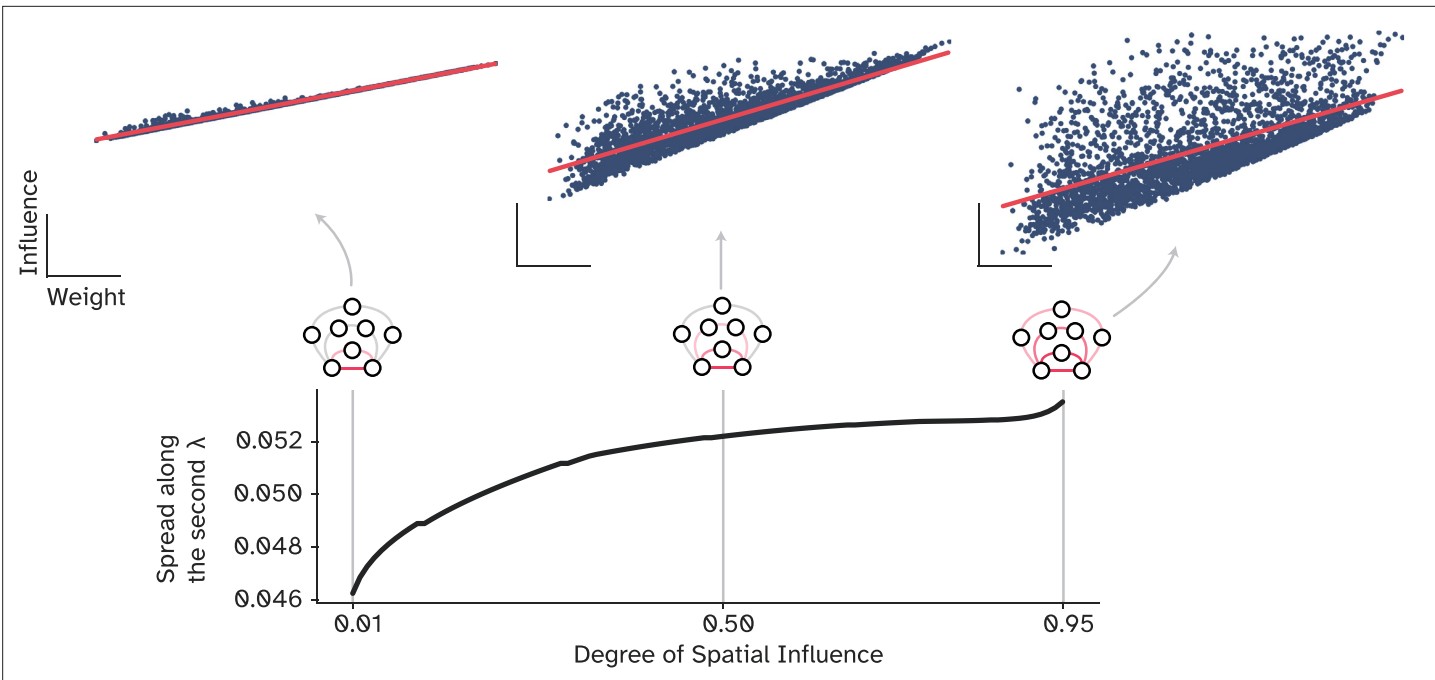

**Figure 10.** How influence decouples from the underlying structural connections. The spread of points along the second principal component, i.e., orthogonal to the regression line, across different values of spatial influence. A very small degree of spatial influence represents a case in which the signal can only reach the neighboring nodes, while higher values indicate influences that extend over longer pathways and reverberate in the network, effectively decoupled from the underlying structural weight.

the topology was held constant. The trend only disappeared when both topology *and* weights were allocated randomly in synthetic random networks (*Figure 8—figure supplement 2C, D, and E*).

We then compared the edge betweenness centrality of every connection with its strength, under both shortest-path and random walk conceptualization. **Shortest-path edge-betweenness centrality (SPEBC)** of a link, measures how many shortest paths in the network go through this specific connection, while **random walk edge-betweenness centrality (RWEBC)** counts the number of times a random walker traversed this link to move from a node to another. While the trend is missing when compared to the connection weight, it is apparent when compared to the influence, suggesting that the effect cannot be captured by how central *individual links* are given these two signaling strategies (*Figure 9A and B*). Together, these findings propose that neither connectivity nor weights *alone* are responsible for the emergence of efficient signal propagation between weakly connected nodes. But the signature of more influence compared to the underlying weight is only observed when central nodes are allocated with weak links.

Lastly, to further investigate this phenomenon, we systematically changed the degree of spatial influence for a SAR model while calculating the spread of data along the second **principal component (PC)**. The first PC captures the main direction of variability between the influence and connection strength, which is the observed strong relationship. However, the second PC captures how much data points are spread orthogonal to the main axis, indicating the amount of which weak links are influential. In other words, this metric measures the variability around the regression line, a small variability indicates a robust prediction of influence from the connection weight, showing their tight coupling. In contrast, large variability approximates how much communication between two nodes was independent of the connection strength, potentially relying on other parallel pathways (*Figure 10*). We found that a steep decay factor leads to low variability, as nodes could only influence their connected neighbors before the signal subsides. A shallower decay rate, on the other hand, resulted in more variability since the signal could traverse longer parallel pathways, echoing in the network and reach the target from multiple fronts (*Figure 10*). This finding provides further evidence for central nodes utilizing not only the direct path but other longer paths to compensate for the weak direct connection strength.

Moreover, we found that these nodes also have both larger average controllability and modal controllability (see *Figure 9—figure supplement 1*), providing insight about their potential role in steering the dynamics of the brain. Average controllability measures the ability of a node to push the global dynamics into easy-to-reach states, while the modal controllability measures the power of a node to push the network towards farther and hard-to-reach states. It has been shown that the nodal strength has a strong positive correlation with average controllability and a strong negative correlation with modal controllability (*Gu et al., 2015*). Our results also support this finding, with the Spearman's rank correlation of nodal influence and average controllability being 0.86 and –0.63 for modal controllability. However, the influential nodes are clustered in the top-right corner of the scatter plot for modal controllability (*Figure 9—figure supplement 1*). This finding suggests that these nodes steer the network to both easy-to-reach and hard-to-reach dynamical states due to their extensive local and long-range connectivity.

Altogether, results from this section propose a mechanism with which hub regions optimally communicate with their distant counterparts, given the structurally weak connections between them. These regions harness their topologically central positions to broadcast their signals not only via one (their direct) connection but other parallel indirect ones to direct the global dynamics of the brain towards different states, even those that are energetically costly to reach. In other words, the weak connection between two 'peripheral' nodes results in weak communication, though, this inefficiency is compensated by central nodes that have access to many pathways to amplify their signal.

## Discussion

In this work, we explored the characteristics of brain communication in a state of optimal signal propagation. We employed a game-theoretical framework to uncover how nodes influence each other optimally, given a set of constraints. We defined these constraints as the network's structural topology and its functional activity, which emerged from three increasingly complex large-scale brain models. These models ranged from a linear model, where the output of each node is the weighted sum of its inputs, via a nonlinear model, in which the node modifies the incoming signals using a Tanh function, to a Hopf model that, relative to the other two models, represents more biophysical realism, such as conductance delay, metastability, and oscillatory dynamics of the brain signals, at the cost of greater computational complexity. Our results indicate that the strength by which nodes influence each other depends less on the local node model and more on the global network structure, permitting us to apply a wide range of graph theoretical metrics that fundamentally relate to the linear model. For instance, comparing putative communication models, we found that the optimal signal propagation in brain networks is better captured by frameworks that conceptualize communication to follow a broadcasting strategy. This means that, although signals travel along multiple parallel pathways, they subside after each processing step and degrade after six to eight steps. Finally, we show that topologically central brain regions, such as precuneus, prefrontal cortex, and anterior cingulate cortex, amplify their influence over remote regions by harnessing such parallel pathways. These results have several implications and caveats that are discussed below.

### Interpreting game-theoretical solutions for influence decomposition

Here, we argue for developing normative formulations of brain signaling and network organization. This approach follows similar studies in synthetic brain networks, where a work also using game theory showed how nodes aiming at maximizing the network's navigability while minimizing their wiring cost organize in structures similar to empirical brain networks (*Gulyás et al., 2015*). Interestingly, related work (*Samoylenko et al., 2023*) addressed the long-standing question of why six degrees of separation exist in social networks (*Travers and Milgram, 1977*). Using normative modeling provided by game theory, the authors found the reason to be a trade-off between a node's aspiration to maximize its centrality, while minimizing its connection maintenance cost (*Samoylenko et al., 2023*). In the present work, we also found that the first eight steps of propagation are sufficient to reconstruct OI, with a peak on the sixth step.

As noted in the introduction, OI is model-agnostic, here, we leveraged this liberty to compare signaling under different models of local dynamics, primarily built upon undirected human connectome. We also included different modalities, e.g., tract tracing in Macaque (see Structural and

functional connectomes under Materials and methods) to confirm the influence of weak connections is not inflated due to imaging limitations (*Figure 8—figure supplement 2A*). The game theoretical formulation of signaling allows for systematic comparison among many combinations of modeling choices and data sources. Nonetheless, in this work, we assumed full observability, i.e., complete empirical knowledge of brain structure and function that is not necessarily practically given. Although a detailed investigation of this issue is needed, mathematical principles behind the method suggest that the framework can isolate the unobserved influences. In these cases, activity of the target node is decomposed such that the influence from the observed sources is precisely mapped, while the unobserved influences form an extra term, capturing anything that is left unaccounted for, see *Algaba et al., 2019a*; *Fakhar et al., 2024c* for more technical details.

More on the mathematical principles, the Shapley value has been shown to converge to a Nash equilibrium in some games, for instance, bargaining in which players have to divide a sum among themselves while maximizing their payoffs, allowing us to draw a connection between this value and the notion of the equilibrium state (*Gul, 1989*). interestingly, this finding has been experimentally replicated, where subjects played a simplified bidding game and frequently settled on division solutions very similar to their Shapley values (*Chessa et al., 2022*). Thus, it is important to note that the equilibrium discussed in the current paper is a game theoretical equilibrium in the solution space (*Figure 2B*) and not necessarily associated with the neural dynamics. Related to this point is the notion of optimality. Here, by optimal influence, we refer to *the optimal solution for decomposing influences, from the cooperative game theoretical perspective, that translates to the non-cooperative game-theoretical definition of optimality as the largest amount of influence that a node can potentially assert on a target given the game constraints*. As depicted in (*Figure 2B*), optimality here refers to the equilibrium point in the solution space. We call it optimal because, as shown before (*Shapley, 1997*), the Shapley value is a unique solution, and in fact the only solution that satisfies three of the four axioms related to fairness (*Algaba et al., 2019b*; *Roth, 1988*) which guarantees stability and dominance of the solution compared to others. Lastly, the Shapley value is being used in the context of interpretability of machine learning features by frameworks such as **SHAP (SHapley Additive exPlanations**; see *Chen et al., 2022a*; *Lundberg and Lee, 2017*). However, what is done in the present work is fundamentally different from SHAP. Although both frameworks compute Shapley values, SHAP does it for input features of machine learning algorithms. In contrast, here we compute the Shapley value of each node of the network, with respect to all other nodes, at every time point. This warrants a different interpretation of the results compared to the SHAP framework.

There is a large body of research aiming at decomposing the amount of influence originating from a set of source nodes on a target node. Some attempts have used information theory, such as causal information (*Ay and Polani, 2008*), and information transfer (*Novelli and Lizier, 2021*), as well as using **Partial Information Decomposition (PID)** (*Ehrlich et al., 2022*; *Gutknecht et al., 2021*; *Luppi et al., 2024*). In PID, the information transferred can be further decomposed into synergistic, redundant and unique contributions. In the current work, we deliberately used the wording of influence instead of information, since a formal link between these two properties still needs to be established (but see *Ay et al., 2019* for an example of equating them). Our framework, however, does not measure the amount of information transferred as defined by information theory; it also does not explicitly decompose the influence into subparts. What it does is to characterize the amount of influence one node has over another at any time points, given a modeling paradigm (see *Figure 7—figure supplement 1*). This influence embeds higher order interactions, such as synergistic and redundant interactions, but eventually summarizes them into one value per time point, describing the average contribution of a node to its targets given its contribution to all possible coalitions of sources. Therefore, we believe that our framework complements the rich body of previous research by providing another level of rigorous granularity, specifically, time-resolved exact influence. Further research is needed to establish where the two perspectives of game theory and information theory converge, and how combining them could provide insights about communication in brain networks.

On a related note, due to its reliance on multi-site lesioning, MSA has been used to address the complexity resulting from higher order interactions in the context of causal inference (*Fakhar et al., 2022*; *Fakhar and Hilgetag, 2022*; *Malherbe et al., 2021*; *Zavaglia et al., 2023*). Following the well-established Woodwardian account of causality, that is, the manipulability account (*Woodward, 2005*), MSA at its core is a causal inference framework. The manipulability account of causality (*Woodward,

*2005*) states that 'C is a cause of E, if manipulating C results in tractable changes in E.' Here, we define manipulation as lesioning, as is customary in neuroscience (*Adolphs, 2016*; *Joutsa et al.,* *2023*; *Siddiqi et al., 2022*; *Vaidya et al., 2019*). We then address the problem of 'counterfactuals' by exploring all possible ways that causes can interact at any time-point to produce an effect, leading to one of the most detailed accounts of causality. Therefore, placing our method in the large family of **effective connectivity (EC)** methods appears as a natural next step. However, here we decided to opt out of this comparison in favor of communication models, since they conceptualize brain-wide causal interactions, which involve both direct and indirect interactions, uncovering causal pathways instead (*Ross, 2018*). Expanding causal relationships to pathways of different lengths allows for multiple causes to interact and jointly produce an effect. Moreover, comparing our findings with those from EC approaches would involve considering a host of methods, not limited to but including spectral dynamic causal modeling (*Seguin et al., 2019*), partial correlation analysis, Granger causality, dynamic differential covariance (*Chen et al., 2022b*; *Laasch et al., 2024*), transfer entropy (*Novelli* *and Lizier, 2021*), and so forth. We believe that this worthwhile comparison calls for a separate and extensive analysis that is beyond the scope of the current work.

Finally, our framework is model-agnostic, meaning that the game and its constraints should be defined first. This approach comes with great liberty to explore different dynamical models. However, we acknowledge that this freedom can be a double-edged sword. On the one hand, it allows researchers to investigate a multitude of questions, potentially providing meaningful insight into the mechanism of communication and its role in behavior. On the other hand, it can result in ill-conceived games that lead to misinterpretations. For instance, here the time-resolved influence of all nodes over each other is a three-dimensional array. Conventionally, it would be tempting to average over the time points and produce a time-averaged influence matrix. However, since our models were stationary, the average value of the vector would be highly noisy, providing little insight into 'how much' nodes modulate each other's activity. Instead, we computed the variance to capture this feature, but one might as well compute alternative metrics, such as standard deviation, entropy, energy, where each requires careful interpretation. Moreover, variance alone might be a poor indicator of influence if the network has inhibitory connections, as both biological and **artificial neural networks (ANNs)** tend to have. By contrast, the present work focuses on conventional neuroimaging datasets with non-negative connection weights.

In sum, it is crucial to define every step of the game carefully to avoid misinterpretations. It is also important to keep in mind the provided definitions for optimality, equilibrium and influence to avoid confusion in interpreting the results of the game-theoretical analysis.

## Broadcasting as an optimal signaling strategy

Many previous studies in network neuroscience have relied on a graph theoretical definition of efficiency that is based on the shortest path distance (*Sporns, 2018*). Here, we show that the underlying signal propagation dynamics in large-scale brain networks are likely to follow a broadcasting regime. This finding implies that broadcasting goes beyond signaling along the shortest path but does not downplay its role. In other words, we show that the shortest path is critical, but not the only path along which a signal traverses. The broadcasting mechanism also circumvents a major problem of the shortest path signaling concept, that nodes would have to find the shortest path to every other node in the network without access to global information on the network organization. By broadcasting, nodes simply transmit their signal via their outgoing connections in every direction. And since the signal degrades over every step, nodes topologically closest to the source receive the signal most strongly. Put differently, navigation along the shortest path emerges naturally and without a need for centralized routing strategies, simply because the closest node receives more signal compared to the farther ones. This line of reasoning is compatible with the result depicted in (*Figure 8E*) where closeness centrality predicted the influential nodes with such great accuracy compared to other centrality measures. Additionally, communication models, such as communicability, are thought to be energetically costlier than routing strategies, such as signaling exclusively along the shortest paths (*Seguin et al., 2023*). The reason for this argument is the fact that communicability accounts for paths of all length. However, as shown here, paths longer than seven steps are unlikely to be meaningfully contributing. Supporting this finding, Griffa and colleagues *Griffa et al., 2023* used information theory and found that not only parallel communication is present in the human cortex, but also

pathways longer than five steps are not as effective. These observations suggest that regions balance between robustness and energetic cost of signal transmission by using a handful of parallel pathways. Intriguingly, (*Griffa et al., 2023*) reported that the capacity for parallel communication is larger in the human cortex compared to macaque and mouse, proposing an evolutionary advantage for broadcasting over multiple pathways (*Griffa et al., 2023*). They also found that these pathways are more frequently placed between different functional modules, rather than within them, which aligns with our finding that hub regions harness parallel pathways to compensate for weak structural connections among them.

Another implication of this finding is that metrics such as global efficiency, representing the average shortest-path distance between all nodes, are not ideal for capturing the signaling dynamics of the brain. As Zamora-Lopez and Gilson argue (*Zamora-López and Gilson, 2024*), all such graph theoretical metrics assume some sort of dynamics. It is important to first establish the correct dynamics and then apply the corresponding metrics. Prompted by our findings, we would suggest that, when possible, metrics based on the shortest path distance should be replaced with others based on communicability, LAM, or SAR. For instance, instead of using the average shortest path as a measure of global network efficiency, one may compute the average communicability as *global broadcasting efficiency*. Similarly, closeness centrality may be redefined as the *broadcasting strength* by taking the average node-wise communicability instead of the average node-wise shortest-path distance.

Next, it is not yet clear whether the broadcasted signal is modified along longer chains in the network, or just relayed to boost the transmitted information. If the signal is modified, then it is likely that these pathways serve a computational role. Previous work suggests that within-module communication involves more redundant information compared to communication across modules which is mainly synergistic (*Luppi et al., 2024*; *Luppi et al., 2022*). Interpreting our work in this perspective, information travels along fewer parallel pathways within a functional module to be processed locally, resulting in higher redundancy. However, signals are likely modified along the parallel pathways and combined with incoming signals from other modules, leading to a greater synergy.

Interestingly, the broadcasting conceptualization is argued to be a better model of information propagation in social networks compared to a random-walker-based navigation (*Ghosh et al., 2023*). In neuroscience, the broadcasting model of information propagation bears qualitative resemblance with concepts such as traveling waves of excitation (*Muller et al., 2018*), bridging a gap between communication models and communication via oscillation. Notably, we found a correlation of 0.23 between OI and FC (*Figure 4—figure supplement 1*), that is close to what has been previously reported by comparing **cortico-cortical evoked potentials (CCEPs)** with FC (*Keller et al., 2011*). Moreover, it has been proposed that at least a subset of CCEPs propagate as traveling waves (*Campbell et al., 2023*). Together, these works, and our findings suggest that, first, correlation-based measures such as FC capture the propagation of information in brain networks to a limited extent, and second an approach combining CMs with traveling waves could provide biophysical mechanisms for simple signal propagation models based on broadcasting regime such as communicability. Nonetheless, our framework is grounded in game theory where its fundamental assumption is that nodes aim at maximizing their influence over each other, given the existing constraints. This assumption is well explored using various theoretical frameworks (*Buehlmann and Deco, 2010*; *Bullmore and Sporns, 2012*; *Chklovskii et al., 2002*; *Laughlin and Sejnowski, 2003*; *O'Byrne and Jerbi, 2022*) and remains open to further empirical investigation. Here, we used game theory to mathematically formalize a *theoretical optimum* for communication in brain networks. Our findings then provide a possible mechanism for achieving this optimality through broadcasting. Based on our results, we speculate that, there exist an optimal broadcasting strength that balances robustness of the signal against its metabolic cost. This hypothesis is reminiscent of the concept of brain criticality, which suggests the brain to be positioned in a state in which the information propagates maximally and efficiently (*O'Byrne and Jerbi, 2022*; *Safavi et al., 2024*). Together, we suggest broadcasting to be the possible mechanism with which communication is optimized in brain networks, however, further research directions include investigating whether signaling within brain networks indeed aligns with a game-theoretic definition of optimality. Additionally, if it does, subsequent studies could then examine how deviations from optimal communication contribute to or result from various brain states or neurological and psychiatric disorders.

Finally, we found that the rich-club organization of the human cortex is a major contributor to shaping brain-wide communication dynamics. An inspiring work using normative modeling suggested that hierarchies and rich-club organization are spandrels of modules and hubs (*Rubinov, 2016*). Spandrels are evolutionary features that evolve alongside adaptive phenotypes, while themselves serve no adaptive function. In other words, the rich-club organization is proposed to be the byproduct of modular organization while serving no adaptive function by itself (*Gould and Lewontin, 1979*). Our work provides evidence that, although the rich club organization could have emerged as a byproduct of modular organization, it might not be a purely ornamental spandrel, as the rich club regions represent the most influential regions of the brain (*Figure 8*). Previous works relating rich club regions to signaling also indicated an integrative role of these regions, which have longer temporal receptive fields compared to others, allowing them to integrate more information over time (*Chaudhuri et al., 2015*; *Hasson et al., 2008*). Interestingly, our results depicted in *Figures 8 and 10* support this idea. We found that rich club regions integrate signals over longer and parallel structural pathways, enhancing their broadcasting capacity. Moreover, a recent empirical work identified disturbances in these regions and their broadcasting efficiency in patients suffering from disorders of consciousness (*Panda et al., 2023*). Put together, these findings suggest a causal functional role for the rich-club organization of the human cerebral cortex.

## Role of network dynamics in optimal signaling

Although our work suggests a greater role for the network's structure in shaping the state of optimal signal propagation in the network compared to the specifics of the local node dynamics, this finding does not mean that the local dynamics are irrelevant. We believe that the inequality between the network's structure and dynamics in determining signal flow arises from factors such as the assumed homogeneity of nodes in our network models. Here, nodes are assumed to be identical objects with identical features and dynamics. This assumption certainly does not apply in the brain, and a growing body of network models that incorporate regional differences aim at addressing this issue (*Bazinet et al., 2023*; *Deco et al., 2021*). As mentioned before, our framework is model agnostic, so it is also possible to interrogate networks with heterogeneous models of dynamics. For applying the present approach to heterogeneous networks, we hypothesize that, first, how nodes influence one another further decouples from the structure (*Fakhar et al., 2022*), and second, simple communication models fail to replicate the OI as well as they have done in this work. Thus, further work can shine light on the interplay between the network's structure and its dynamics in shaping OI, since not only heterogeneous network models are biologically more plausible, but also, they can help to investigate how information optimally flows in networks where nodes respond differently to incoming signals.

## Limitations and further directions

A conceptual note is that OI, as a normative framework, describes the landscape of communication dynamics if nodes maximize their influence. As in other game-theoretical models, this assumption might not fully hold. The derived landscape is a prediction that would need to be experimentally tested using in-vivo multi-site lesioning experiments, which is currently not feasible. However, recent work indirectly supports our findings by showing that propagation of electrical stimulation in the human cortex is well captured by communicability (*Seguin et al., 2022a*), which we found to have one of the largest correlations with OI. There are also technical limitations of our framework, specifically its computational cost, since one needs potentially to simulate millions of in-silico lesion combinations for each target node. For instance, in this work, we performed around half a billion lesion simulations for just one game, such as a linear model on the human connectome (see section Game-theoretical Framework in Material and Methods). Performing such an intensive analysis is not possible on a simple laptop. Therefore, due to its brute-force approach, our framework provides rigorous results at the expense of high computational effort, apart from the fact that it is yet to be used in-vivo (but see *Zavaglia and Hilgetag, 2016*).

Finally, we focused on the influence of nodes on each other without relating them to behavior or cognitive function. Moreover, we did not investigate how this influence landscape changes due to brain disorders, aging, and other factors. Future work may take on these challenges, for instance, by tracking the dynamics of OI during neurodegenerative disease progression.

# Materials and methods

In this study, we used multiple Python libraries, including MSApy (*Fakhar, 2021*), Neurolib (*Cakan et al., 2023*), Netneurotools, BCTpy (*Rubinov and Sporns, 2010*), nctpy (*Parkes et al., 2022*), and Networkx (*Hagberg et al., 2008*). The Jupyter notebooks and Python functions to reproduce this work are available at the GitHub repository: https://github.com/kuffmode/OI-and-CMs (copy archived at *Fakhar and Dixit, 2024a*).

The simulation results are available at: https://zenodo.org/records/10849223.

Additionally, a dedicated user-friendly Python library to compute OI and some communication models is developed and can be found at: https://github.com/kuffmode/YANAT (copy archived at *Fakhar and Dixit, 2024b*).

All connectomes are available as a part of the Netneurotools library: https://github.com/netneurolab/netneurotools (copy archived at *Bazinet et al., 2025*).

## Structural and functional connectomes

We analyzed three publicly available connectomes of human, macaque, and mouse to address differences in invasive versus non-invasive imaging modalities and reconstruction techniques.

### Human connectome

Structural and functional connectivity datasets were acquired from a cohort of 70 healthy subjects at the Lausanne University Hospital in Switzerland, and the details are described elsewhere (*Griffa et al., 2019*; *Shafiei et al., 2020*; *Shafiei et al., 2019*). Briefly, the group comprised individuals with an average age of 28.8 y (standard deviation 9.1 y), including 27 females, and all were scanned using a 3 Tesla MRI machine. A deterministic streamline tractography was employed to create individual SC matrices from each participant's **diffusion spectrum imaging (DSI)** data, which were recorded at five distinct levels of brain parcellation. However, here we have used only one with relatively high-resolution (219 cortical regions). The strength of each structural connection within these matrices was approximated by the density of fiber tracts.

To mitigate the size disparities between brain regions and counteract the inherent preference for longer fiber tracts in the tractography technique, a normalization procedure was applied. This involved adjusting the streamline counts by the average surface areas of the connected regions and the mean streamline lengths. The culmination of this process was the formation of a unified, group-average weighted SC matrix, which was derived using a consensus strategy that ensured the individual edge length distributions were conserved (*Betzel et al., 2019*; *Griffa et al., 2019*).

Functional brain data were gathered from the same group of subjects through **resting-state functional MRI (rs-fMRI)** scans conducted with open eyes. The initial processing steps of this functional data included adjustments for several physiological variables, notably the influence of white matter, cerebrospinal fluid, and motion artifacts that comprise three axes of translation and rotation as determined by rigid body co-registration. Following these corrections, the **blood oxygen level-dependent (BOLD)** time series were subjected to a low pass filtering process, utilizing a temporal Gaussian filter set to a full-width half maximum of 1.92 s. The analysis excluded the first four temporal points from each scan and further data refinement was achieved through high-motion frame censoring, a method detailed in Power and colleagues (*Power et al., 2012*).

For the assessment of FC, the study calculated zero-lag Pearson correlation coefficients, which measured the functional relationships between pairs of brain regions in each individual's rs-fMRI time series. Lastly, the group-average FC matrix was computed as the mean of these individual pairwise connectivity values, which includes negative values as well.

### Macaque connectome

The connectivity matrix for the macaque was derived from retrograde tract-tracing experiments and is provided by *Markov et al., 2013*. These experiments involved the use of fluorescent tracers in a group of 28 macaque monkeys. The projections that were reconstructed from these experiments were mapped according to a division of 91 cortical areas. This division was based on a combination of histological examinations and atlas-based references. In each tract-tracing experiment, Markov and colleagues quantified the number of neurons that were labeled in each of these 91 areas. The count of

labeled neurons was then adjusted by subtracting the number of neurons that were native to the site of tracer injection. The outcome of this process was a 29×91 matrix, which represents the connection weights extending from each injection site to other regions of the brain. Here, we focused on the subset of this matrix that describes the weighted and directed connections between 29 cortical areas, leading to a 29×29 SC matrix.

## Mouse connectome

The connectivity matrix for the mouse was compiled using tract-tracing data openly accessible from the Allen Institute Mouse Brain Connectivity Atlas (*Oh et al., 2014*). Briefly, the process involved injecting anterograde recombinant adeno-associated viral tracers into designated areas within the right hemisphere of mouse brains. Three weeks following the injection, during which the viral tracer projections developed, the brains were extracted for reconstruction. These reconstructions were then standardized and aligned with the Allen Reference Atlas' common coordinate framework.

Nodes in the network were identified based on a specialized parcellation derived from the Allen Developing Mouse Brain Atlas. This parcellation initially included 65 areas in each hemisphere, but 9 areas were excluded as they did not participate in any tract-tracing experiments. Consequently, the network that was analyzed constituted 112 regions (*Rubinov et al., 2015*). Edges represent axonal projections between different areas, and they were quantified as normalized connection densities. Specifically, this measure represents the number of connections per unit volume from a source area to a target area.

## Large-scale computational models of the brain dynamics

In this work, we employed three models with increasing biological fidelity. The most abstract and simplistic model is the linear model, where the output of a node is a weighted sum of its inputs. The nonlinear model extends the linear model by applying a nonlinear transformation to the input (here, $\tanh(.)$) And lastly, the neural mass model describes nodes as Stuart-Landau oscillators. Below are the details of each model.

### Linear and nonlinear models

The linear model we used, also known as the multivariate Ornstein-Uhlenbeck process, follows the continuous dynamical system equations described in *Fernández Galán, 2008*.

$$\tau \frac{dx_i}{dt} = -x_i(t) + g \sum_{j=1}^{N} W_{ij} x_j(t) + \sigma_{in} \boldsymbol{\xi} \tag{1}$$

Independent Gaussian noise $\boldsymbol{\xi}$ with mean zero and unit variance were presented to each node. Here, $\tau$ denotes the relaxation time of the node, which was set to 0.02. It should be noted that, in practice, the adjacency matrix is normalized to ensure that its largest eigenvalue $|\lambda_{max}| = 1$. Consequently, the practical decay rate aligns with the product of the relaxation time, $\tau$, and the maximum eigenvalue, $\lambda_{max}$. $\sigma_{in}$ denotes the strength of the noise and was set to 0.05 in all experiments. Given the empirical functional connectivity matrices, we optimized the coupling parameter, $g$, to maximize correlation between empirical and simulated FCs. With our modeling setting, $g = 0.74$ demonstrated the best fit, with a correlation of 0.23. The nonlinear model modifies the linear one simply by adding a nonlinear transformation and a larger input noise of 2 instead of 0.05 to go beyond the linear part of the function, leading to the equation below:

$$\tau \frac{dx_i}{dt} = -x_i(t) + \tanh\left( g \sum_{j=1}^{N} W_{ij} x_j(t) + \sigma_{in} \xi \right) \tag{2}$$

### Neural mass model

Furthermore, we explored SC networks governed by Hopf/Stuart-Landau equations (*Deco et al., 2017*). This model, recognized as the canonical approach for examining the shift from noisy to oscillatory dynamics, elucidates the behavior of a nonlinear oscillating system near the Hopf bifurcation. In essence, the dynamics of each network node are captured by the following complex equation:

$$\frac{dz_i}{dt} = \left(a_i + j\omega_i - |z_i|^2\right) z_i + \sigma_{in}\xi_i \tag{3}$$

Where $z_i = x_i + jy_i$ is a complex representation of the node state, with $\xi_i(t)$ representing Gaussian noise characterized by a standard deviation $\sigma_{in} = 0.05$. This system undergoes a supercritical bifurcation at $a_i = 0$, where it shifts from a stable fixed point (i.e. $z_i = 0$) to a limit cycle oscillation with frequency $f_i = \omega_i/2\pi$.

By decomposing the complex state into its Cartesian components and incorporating the influence of other nodes in the input of each node, we derived a set of coupled equations (**Equation 4**) to govern the entire brain dynamics. This formulation enables the representation of the influence of nodes on each other's temporal state through a diffusive interaction, where $g$ serves as a global coupling factor.

$$\frac{dx_i}{dt} \left(a_i - x_i^2 - y_i^2\right) x_i - \omega_i y_i + g\sum_i W_{ij}\left(x_j - x_i\right) \sigma_{in}\xi_i$$
$$\frac{dy_i}{dt} \left(a_i - x_i^2 - y_i^2\right) y_i + \omega_i x_i + g\sum_i W_{ij}\left(y_j - y_i\right) \sigma_{in}\xi_i \tag{4}$$

Similar to the other two models, the global dynamics of the network are influenced by model's parameters. Here, we precisely tuned the global coupling, $g$ and the bifurcation parameter, $a_i$, to match the empirical FC. Setting g at 5.6 and $a_i = 0.15$ for all nodes, yields a correlation of 0.35. Other parameters, as detailed in the Neurolib library, remain unaltered as follows. The model features a diffusive coupling type, a signal velocity of 20.0 m/s, a global coupling strength set at 5.8, a 5.0 ms Ornstein-Uhlenbeck timescale, Ornstein-Uhlenbeck noise intensity fixed at 0.05 mV/ms/sqrt(ms), a mean value of Ornstein-Uhlenbeck process maintained at 0.0 mV/ms, a Hopf bifurcation parameter set to 0.15, and an oscillator frequency of 32 Hz. We collected state variables, $X$, representing signals acquired from various brain regions for subsequent analysis.

## Game-theoretical framework

MSA is built upon Shapley values, which quantify a player's *fair* share of a collectively generated outcome (see **Figure 2**). Generally, Shapley values are calculated by adding a player to all possible coalitions and observing the value they bring to the coalition (**Keinan et al., 2004b**). Formally, the contribution of a player $i$ to a coalition $S$ is given by:

$$\Delta_i(S) = v\left(S \cup \{i\}\right) - v(S)$$

Where $\Delta_i(S)$ represents the value of coalition $S$. The Shapley value $\gamma_i$ is the average of these contributions across all permutations $R$ of the player set:

$$\gamma_i(N, v) = \frac{1}{n!} \sum_{R \in \mathfrak{R}} \Delta_i\left(S_i(R)\right)$$

where $\mathfrak{R}$ is the set of all permutations, and $S_i(R)$ is the coalition formed by ordering $R$ up to player $i$. However, due to the computational complexity of calculating Shapley values for large sets, MSA relies on an unbiased estimator by sampling permutations. We employed a sample size $m \ll N!$, generating $N \times m$ unique permutations for every target region, where $N$ is the number of source regions.

In this work, we iterated this process over every node (target node) by systematically lesioning other nodes (source nodes) and tracking the time-resolved difference between when the sources were lesioned and when they were not. Lesioning was modeled by setting the incoming and outgoing connections of sources to zero. For every source node, we sampled m=1,000 permutations. However, we confirmed that the algorithm converged by comparing it against a larger sample size of m=10,000, which resulted in a correlation coefficient of 1.0 (**Figure 2—figure supplement 1**). We also ran the analysis for 10 repetitions and averaged the resulting influence matrices. The final matrix has a shape of (number of regions × number of regions × simulation time) that we reduced to a two-dimensional matrix (number of regions ×number of regions) by taking the variance of each influence profile. Altogether, each of the experiments on the human connectome resulted in roughly 480 million in-silico lesions (219 targets × 219 sources × 1,000 combination of lesions per source for each target ×10

**Table 1.** Communication models and measures.

| Models | Measures* | References |
|---|---|---|
| Shortest Path Efficiency Floyd-Warshall algorithm | $SPE = \frac{1}{\Lambda^*}$ <br> $\Lambda_{ij}^{\Omega^*} = L_{iu_1} + \cdots + L_{u_k j}$ <br> $\Omega_{ij}^* = \min\limits_{\Omega ij=\{i,u_1,\ldots,u_k,j\}} \frac{1}{\Sigma_{l=1}^k w_{u_l u_{l+1}}}$ | *Latora and Marchiori, 2001; Seguin et al., 2020* |
| Navigation Efficiency | $NE = \frac{1}{\Lambda}$ <br> $\Lambda_{ij}^{\Omega} = L_{iu_1} + \cdots + L_{u_k j}$ <br> $L_{st} = \min\limits_{(s,s+1)\in V} \|D_{st}\|_2^2$ | *Seguin et al., 2020; Seguin et al., 2018* |
| Diffusion Efficiency | $DE = \frac{\Sigma_i \Sigma_j \frac{1}{t_{ij}}}{N(N-1)},\ i \neq j$ <br> $t_{ij} = \frac{z_{jj} - z_{ij}}{\omega_j},\ i \neq j$ <br> $Z = [z_{ij}] = (\mathbb{1} - \hat{W} + \hat{\omega})^{-1}$ <br> $\hat{W} = [\hat{w}_{ij}],\ \hat{w}_{ij} = \frac{W_{ij}}{\Sigma_{u=1}^N W_{iu}}$ <br> $\omega = [\omega_1, \ldots, \omega_N]^T,\ \omega \hat{W}^T = \lambda\omega.$ <br> $\hat{\omega} = [\omega, \ldots, \omega]$ | *Goñi et al., 2013; Seguin et al., 2020* |
| Search Information | $SI = -\log_2(\Pi)$ <br> $\Pi_{\Omega_{ij}} = \hat{w}_{iu} \times \cdots \times \hat{w}_{ij}$ | *Seguin et al., 2019; Seguin et al., 2020* |
| Communicability | $CO = [co_{ij}]$ <br> $co_{ij} = \Sigma_{k=0}^{\infty} \frac{(\hat{W})_{ij}^k}{k!} = e^{\hat{W}}$ | *Chen et al., 2022b; Estrada and Hatano, 2008* |
| Scaled Communicability | $SCO = [sco_{ij}]$ <br> $sco_{ij} = \Sigma_{k=0}^{\infty} \frac{(\alpha\hat{W})_{ij}^k}{k!} = e^{\alpha\hat{W}}$ | *Ghosh et al., 2023; Messé et al., 2014* |
| Linear Attenuation | $LAM = (\mathbb{1} - \alpha\hat{W})^{-1}$ | *Goñi et al., 2013* |
| Spatial Autoregressive | $SAR = (\mathbb{1} - \alpha\hat{W})^{-1}(\mathbb{1} - \alpha\hat{W})^{-T}$ | *Zamora-López et al., 2016* |

*Throughout the table, $W \in \mathbb{R}^{N \times N}$ is the adjacency matrix corresponding to a given graph, $G = (V, E)$ with $N$ vertices (nodes). $W_{ij}$ signifies the connection strength from vertex i to vertex j. Anatomical connections are indicated by $W_{ij} > 0$ for connected region pairs and $W_{ij} = 0$ for unconnected pairs. $L$ denotes the matrix of path lengths, defined as the reciprocal of $W$, such that $L = \frac{1}{W}$ with $L_{ij}$ as the path length between two connected nodes, i and j. Further, $\Omega_{ij} = \{u_1, u_2, \ldots, u_k\}$ is the sequence of nodes visited along the shortest path between nodes i and j. Given the spatial organization of nodes in empirical structural connectivity, $D_{st}$ includes the Euclidean distance between source, s, and destination, t nodes in finding optimal paths. The term $\hat{W}$ signifies the normalized adjacency matrix, where each element $\hat{w}_{ij}$ is obtained by dividing the original connection strength, $w_{ij}$ by the sum of all connection strengths in the corresponding row of the matrix $W$. $\mathbb{1}$ is the identity matrix, $T$ stands for transposed inverse matrix, and $\alpha < \frac{1}{\lambda_{max}}$ where $\lambda_{max}$ is the spectral radius of the adjacency matrix, $W$. $\lambda$ denotes the eigenvalues of a given matrix.

trials). Due to the computational implausibility of the neural mass model, we ran the analysis only once instead of 10 repetitions, thus 48 million lesions. Together with all control analyses, the total number of in-silico lesions for the human connectome amounts to 7200 million:

480 m for the linear model, 480 m for the nonlinear, 48 m for the Hopf, 480 m for the topology-shuffled null model, 480 m for the weight-shuffled null model, 4,800 m for the case where we used 10,000 lesion combinations per source for each target, and 480 m for a test case where the coupling was set to zero. All experiments were conducted on a high-performance computing facility provided by the Institute of Computational Neuroscience, University Hospital of Hamburg.

## Communication models and measures

In this section, we introduce the communication models and measures employed to analyze the dynamics of information flow within the studied network. The corresponding formulas and procedures for computation are presented in *Table 1*.

The structural connectivity matrix $W \in R^{N \times N}$ denotes the strength of pairwise connections between $N$ brain regions. We define the connection length matrix $L = 1/W$, where $L_{ij}$ is the travel cost between regions $i$ and $j$. This conversion from connection weights to lengths is required for network communication models that optimize for minimal transmission cost of signals, e.g., shortest path efficiency.

### Shortest path efficiency

Shortest Path Efficiency measures the effectiveness of communication along the most direct route between two nodes in a network. Here, the Floyd-Warshall algorithm was employed to determine the sequence of regions $\Omega_{ij} = \{i, u, \dots, v, j\}$ that minimizes the total transmission cost for signals traveling between regions $i$ and $j$. This cost, denoted by $\Lambda_{ij}^*$ is defined as the sum $L_{iu} + \cdots + L_{vj}$. Subsequently, the shortest path efficiency (SPE) between two regions is quantified as the reciprocal of the minimum transmission cost, expressed as $SPE_{ij} = 1/\Lambda_{ij}^*$ as detailed in *Latora and Marchiori, 2001*.

### Navigation efficiency

Navigation within the network employs a greedy protocol that assumes signal transmission to minimize the inter-regional Euclidean distance. The process involves iterative progression from a source region to a target region $j$. At each step, the neighbor spatially closest to $j$ is chosen as the next node in the path (*Seguin et al., 2018*). This sequence continues until the target is reached, marking successful navigation, or a previously visited node is encountered, indicating a failure in navigation. The cumulative length of a successful path is denoted by $\Lambda_{ij} \Lambda_{ij} = L_{iu} + \cdots + L_{vj}$, where $\Omega_{ij} = \{i, u, \dots, v, j\}$ represents the sequence of nodes traversed. If navigation fails, $\Lambda_{ij} \Lambda_{ij}$ is set to infinity. Navigation efficiency is thus defined as the reciprocal of the traversed path length.

### Diffusion efficiency

The Diffusion Efficiency (DE) model characterizes signaling through the lens of random walks. The computation of DE involves the utilization of the transition matrix, $P$ within a Markov chain process unfolding on the connection weight matrix, $W$. Specifically, it considers the probability $p_{ij}$ that a simple random walker at node $i$ will advance to node $j$ and the mean first passage time, $t_{ij}$ which quantifies the expected number of intermediate regions visited in a random walk. To elaborate on the computation procedure, please refer to *Table 1*. Additionally, for a more in-depth theoretical understanding, consult references (*Goñi et al., 2013*; *Zhou, 2003*).

### Search information

Search information is a metric that evaluates the required amount of information to push a random walker towards the shortest path, thereby reflecting the accessibility of efficient communication routes under a diffusive model. Computing SI involves finding the above-mentioned shortest path, $\Omega \omega_{ij}$ between two regions, and the probability, $\Pi_{ij}$, that a random walker will accidentally traverse from region $i$ to region $j$ following this path. This probability is computed using the transition matrix, $P$ within a Markov chain process unfolding on the connection weight matrix. *Table 1* summarizes the computation procedure.

### Communicability and scaled communicability

The measure of communicability between nodes, $i$ and $j$, is expressed as the weighted sum of the overall pathways connecting them, where each walk's contribution is weighted proportionally to the inverse of its length—signifying the number of connections traversed. In practice, before computing communicability, nonbinary connection weight matrices are commonly normalized. This normalization step is employed to diminish the impact of highly influential nodes with substantial strength. Refer to *Table 1* for the computation procedure and references (*Estrada and Hatano, 2008*; *Zamora-López and Gilson, 2024*) for additional theoretical details. We also reported the scaled communicability measure, which introduces the modulating parameter, $\alpha$ to control the decay rate.

## Linear attenuation model

The Linear attenuation model (LAM) follows the same line of reasoning as communicability, that information propagation can be represented as a weighted sum of all walks in the network. The difference lies in how walk-lengths are discounted. In contrast to communicability, where the sum of walks is estimated through the exponentiation of the adjacency matrix, Katz introduced an attenuation factor, $\alpha$, to reduce the influence over every step monotonically. Katz proposed a closed-form expression, given by $(I - \alpha W)^{-1}$ (*Katz, 1953*). It is important to note that the convergence condition in this context is $0 < \alpha < 1/\lambda_{max}$, where $\lambda_{max}$ represents the spectral radius of the adjacency matrix, $W$. Prior to computing LAM, the adjacency matrix undergoes a normalization process, similar to the one applied in the computation of communicability.

## Spatial autoregressive model

Moreover, we incorporated the spatial autoregressive (SAR) model into our analysis. This model, characterized as a generic representation of diffuse processes on networks, is intricately linked to the distribution of paths within the network (*Betzel and Bassett, 2018*; *Messé et al., 2014*; *Tononi et al., 1994*; *Zamora-López et al., 2016*). In this model, the fluctuating signals at each node are interconnected through a system of structural equations, expressing each signal as a linear function of others. This interconnection is influenced by a global coupling strength factor denoted as $\alpha$. Importantly, when the network experiences input noise, the SAR model predicts that each signal follows a multivariate normal distribution. This prediction is mathematically represented by the covariance matrix $(I - \alpha \widehat{W})^{-1}(I - \alpha \widehat{W})^{-T}$ The term $W$ denotes the normalized adjacency matrix, where each element $w_{ij}$ is obtained by dividing the original connection strength, $w_{ij}$ by the sum of all connection strengths in the corresponding row of the matrix $W$. Here, $I$ is the identity matrix and $-T$ stands for the transposed inverse matrix.

## Null and simulated network models

We employed two simulated weighted networks and two null models as described below:

### Erdős-Rényi model

An Erdős-Rényi random graph, denoted as $G(N, P)$, was generated with 100 nodes, where $p = 0.35$ represents the probability that any two distinct nodes are connected by an edge. The graph was constructed by iterating over all possible pairs of nodes and connecting them with an edge with the probability $p$. This resulted in a binary adjacency matrix $W$ where $W_{ij}$ is 1 if nodes $i$ and $j$ are connected and 0 otherwise. After generating the binary structure of the graph, weights were assigned to the edges by sampling from a log-normal distribution. The weight $W_{ij}$ for each edge in the graph was drawn from $ln\mathfrak{N}(\mu, \sigma^2)$ where $\mu = 1$ and $\sigma = 0.5$. The log-normal distribution was chosen for its property of providing a multiplicative effect, appropriate for modeling local dynamical systems coupled in a network.

### Barabási-Albert model

The Barabási-Albert model was used to generate synthetic scale-free networks through a preferential attachment mechanism. As with the Erdős-Rényi model, we generated a network with 100 nodes. Each new node added to the network creates $m = 20$ edges to existing nodes, with a preference for nodes that already have a higher degree, thus simulating the 'rich get richer' phenomenon and producing hubs in the network. In this Barabási-Albert model, the probability that a new node will be connected to a node $i$ depends on the degree $k_i$, according to the rule:

$$\Pi\left(k_i\right) = \frac{k_i}{\Sigma_j k_i}$$

Where $\Sigma_j k_i$ is the sum of degrees of all existing nodes at the time of attachment. Similar to the Erdős-Rényi model, the edges of the resulting binary network were weighted by sampling from a log-normal distribution with parameters $\mu = 1$ and $\sigma = 0.5$. Moreover, to model weakly connected hubs, all hubs in the network were fully connected with weights assigned from a separate log-normal distribution characterized by both mean and standard deviation equal to 0.1. This simulates the scenario

where hubs have additional connections among themselves with relatively smaller weights, representing the observed weaker links of the rich-club in the brain.

## Topology-conserving null model

To investigate the impact of connection strength on network dynamics while maintaining the underlying topology, a weight shuffling procedure was implemented. The adjacency matrix was kept unaltered to preserve the original connectivity pattern, while the weights were redistributed to randomize the strength of the connections. This process was defined as follows:

> Given a weighted adjacency matrix, $W$, with fixed topology, the set of non-zero weights corresponding to the edges in $W$ was permuted. The permutation was performed in such a manner that each edge received a new weight from the set, ensuring that the sum of the weights and the overall weight distribution remained unchanged. This shuffling procedure was iterated 10 times, yielding 10 distinct weighted networks with identical topologies but randomized weight configurations.

## Weight-conserving null model

To study the impact of topology, we used a degree-, weight-, and strength-preserving model introduced by *Rubinov and Sporns, 2011* in which the connectivity was shuffled while the total degree and strength of the nodes were preserved. We chose the default parameters provided by the BCTpy toolbox, however, the correlation between the strength sequence of pre- and post-rewired network was $r$=0.97 that implies a robust rewiring. As per *Rubinov and Sporns, 2011*, the rewiring algorithm consists of two steps:

1. The network is randomized while maintaining each node's degree using a connection-switching algorithm, here Maslov-Sneppen (*Maslov and Sneppen, 2002*). In this process, connections are adjusted by switching pairs of edges in a way that preserves the sum of weights for the involved nodes.
2. Original network weights are then reassigned to the randomized network. This reassignment is done by first ranking all weights by magnitude and then associating them with the network's connections to approximate the original distribution of weights. During this process, weights are iteratively matched and re-ranked until all connections in the new network closely resemble the original positive and negative strengths.

## **Multivariate statistical model**

We conducted two sets of Lasso regularized multivariate regression models (*Figure 6B and C*). In one, we systematically explored the dynamics of feature importance in predicting the optimal signal propagation and using the other set, we investigated the number of steps needed to predict the optimal signal propagation.

For the first set, we constructed a feature matrix $X$ encompassing all communication measures, topological and geodesic distance measures, FC, structural weights, and fiber length between regions.

A range of 50 $\alpha$ (see *Table 1*) values from 0 to 1, delineating the degree of spatial influence, was explored. For each $\alpha$, the SAR and LAM models were computed and subsequently integrated into the feature matrix $X$. A total of 25 repetition runs were performed for each $\alpha$, wherein the dataset was randomly partitioned into training and testing subsets of proportion 0.7–0.3 using a 10-folds cross validation approach. The training set was subjected to standard scaling before model fitting. The statistical model was assessed through the coefficient of determination $R^2$. The absolute values of the regression coefficients obtained from the model were normalized to sum to 100, reflecting the relative contribution of each feature to the model. These contributions were recorded for each $\alpha$ and averaged over all trials to ascertain the consistent predictors.

For the second set, a range of 15 steps were generated by first raising the structural connectivity matrix to the power of 0–15 and then discounting the longer paths according to the optimal discount factor of the LAM model α. For each step, 100 repetitions were performed following the same cross-validation, evaluation, and feature importance ranking introduced above.

**Table 2.** Graph-theoretical measures.

| Types | Measures* | References |
|---|---|---|
| Closeness Centrality | $C_C(i) = \frac{N-1}{\Sigma_{j=1, j\neq i}^N d_{ij}}$ | *Oldham et al., 2019* |
| Eigenvector Centrality | $C_E(i) = \|v_i\|, \ Wv = \lambda v$ | *Oldham et al., 2019* |
| Node Betweenness Centrality | $C_B(i) = \Sigma_{s\neq i \neq t} \frac{\sigma_{st}(i)}{\sigma_{st}}$ | *Oldham et al., 2019*; *Zuo et al., 2012* |
| Edge Betweenness Centrality | $C_{B_{edge}}(e) = \Sigma_{s\neq t} \frac{\sigma_{st}(e)}{\sigma_{st}}$ | *Oldham et al., 2019*; *Zuo et al., 2012* |
| Random Walk Centrality | $T = D - A, \ D_{ii} = \Sigma_{j=1}^N W_{ij}$ <br> $V_{st}^i = T_{is} - T_{it}$ <br> $I_{st}^i = \frac{1}{2}\Sigma_j W_{ij}\|V_{st}^i - V_{st}^j\|$ <br> $C_{RW}(i) = \frac{\Sigma_{s<t} I_{st}^i}{\frac{1}{2}N(N-1)}$ | *Newman, 2005* |
| Average Controllability | $W = U^* TU$ <br> $M = (U \circ U)^T$ <br> $v = diag(T)$ <br> $P = diag(1 - vv^T)$ <br> $P = [P_{\text{diag}}, \ldots, P_{\text{diag}}]$ <br> $C_{avg} = \sum\left(\frac{M}{P}\right)$ | *Gu et al., 2015* |
| Modal Controllability | $W = U^* TU$ <br> $v = [v_1, \ldots, v_N] = diag(T)$ <br> $C_M(i) = \Sigma_{j=1}^N U_{ij}^2 (1 - v_j^2)$ | *Gu et al., 2015* |

*Throughout the table, $W \epsilon \mathbb{R}^{N \times N}$ is the adjacency matrix corresponding to a given graph, $G = (V, E)$ with vertices (nodes) $W_{ij}$ signifies the connection strength from vertex i to vertex j Anatomical connections are indicated by $W_{ij} > 0$ for connected region pairs and for $W_{ij} = 0$ unconnected pairs. $d_{st}$ denotes the shortest path distance between s nodes and, t and $\sigma_{st}$ is the total number of shortest paths from node s to node t and $\sigma_{st}(i)$ is the number of those paths that pass through node i.In edge betweenness centrality, $\sigma_{st}(e)$ stands for the number of those paths that pass through a given edge, e. Further, v refers to the left eigenvector associated with the eigenvalue of $\lambda$ maximum modulus. In computing controllability measures, we used Schur decomposition to find the unitary matrix, $U$, and the upper triangular matrix, $T$, to express the adjacency matrix, $W$. Further, o denotes element-wise multiplication, * represents the conjugate transpose, and $diag(.)$ extracts the diagonal elements.
†

## Graph-theoretical measures

Several graph-theoretical measures were employed in this study that are summarized in *Table 2* and briefly explained here.

### Closeness centrality

Closeness centrality is a measure reflecting the average shortest path length from a given node to all other nodes in the network. It quantifies how 'close' a node is to all other nodes, which can indicate the node's efficiency in spreading information through the network. As with the SPE, the shortest path distance was calculated using the Floyd-Warshall algorithm.

### Eigenvector centrality

Eigenvector centrality extends the concept of centrality by not only considering the number of connections a node has, but also the centrality of its neighbors (*Oldham et al., 2019*; *Zuo et al.,*

*2012*). It assigns relative scores to all nodes in the network, based on the principle that connections to high-scoring nodes contribute more to the score of the node than equal connections to low-scoring nodes. As detailed in *Table 2*, this quantity at the site of $i$-th node is calculated as the $i$-th component of the eigenvector of the adjacency matrix.

## Shortest path node and edge betweenness centrality

Shortest path betweenness centrality measures the extent to which a node lies on the shortest paths between other nodes in the network. It captures the influence of a node over the flow of information in the network by identifying nodes that frequently act as bridges along the shortest paths between other nodes (*Freeman, 1977*). Subsequently, the shortest path *edge* betweenness centrality quantifies the number of times an edge acts as a bridge along the shortest path between two nodes. It reflects the importance of an edge in facilitating communication within the network. To elaborate on the computation of these metrics, please refer to *Table 2*.

## Random-walk node and edge betweenness centrality

Random walk betweenness centrality, also known as current flow betweenness centrality, is also a measure that quantifies the node's role in facilitating information flow in the network (*Newman, 2005*). However, unlike shortest path betweenness centrality, which only considers the shortest paths, random walk centrality is based on the probability that a random walk starting at a source node will pass through a given node before reaching the destination node. For detailed information on the computation of Random-walk Node and Edge Betweenness Centrality based on the current flow notion, refer to *Table 2*.

## Average and modal controllability

Average controllability measures the ability of a node to steer the system into many easy-to-reach states, given a discrete dynamical system. It is particularly useful in understanding the ease with which any state of the system can be reached from a given initial state. Similarly, modal controllability represents the ability of a node to steer the system into farther and hard-to-reach states. In computing the controllability measure, we used Schur decomposition to find the unitary matrix, $U$, and the upper triangular matrix, $T$, to express the adjacency matrix, $W$. The computation procedure is summarized in *Table 2*.

## Acknowledgements

The funding is gratefully acknowledged: KF: German Research Foundation (DFG)-SFB 936–178316478 A1; TRR169-A2; SPP 2041/GO 2888/2–2; and the Templeton World Charity Foundation, Inc under grant TWCF-2022–30510. FH: DFG TRR169-A2. SD: SFB 936–178316478 A1. AM: SFB 936–178316478 A1. BM: the Natural Sciences and Engineering Research Council of Canada (NSERC Discovery Grant RGPIN #017–04265). The Brain Canada Future Leaders Fund and the Canadian Institutes of Health Research (CIHR). CS: N/A; GZ: N/A; CH: SFB 936–178316478 A1; TRR169-A2; SFB 1461 /A4; SPP 2041/HI 1286/7–1, the Human Brain Project, EU (SGA2, SGA3).

## Additional information

### Funding

| Funder | Grant reference number | Author |
| --- | --- | --- |
| Deutsche Forschungsgemeinschaft | SFB 936-178316478 | Kayson Fakhar<br>Shrey Dixit<br>Arnaud Messé<br>Claus C Hilgetag |
| Deutsche Forschungsgemeinschaft | TRR169 | Kayson Fakhar<br>Fatemeh Hadaeghi<br>Claus C Hilgetag |

| Funder | Grant reference number | Author |
|---|---|---|
| Deutsche Forschungsgemeinschaft | SPP 2041/GO 2888/2-2 | Kayson Fakhar |
| Deutsche Forschungsgemeinschaft | SPP 2041/HI 1286/7-1 | Claus C Hilgetag |
| Deutsche Forschungsgemeinschaft | SFB 1461 | Claus C Hilgetag |
| Human Brain Project | EU (SGA2) | Claus C Hilgetag |
| Natural Sciences and Engineering Research Council of Canada | Discovery Grant RGPIN #017-04265 | Bratislav Misic |
| Brain Canada | Future Leaders Fund | Bratislav Misic |
| Canadian Institutes of Health Research | | Bratislav Misic |
| Human Brain Project | EU(SGA3) | Claus C Hilgetag |
| Templeton World Charity Foundation | 10.54224/30510 | Kayson Fakhar |

The funders had no role in study design, data collection and interpretation, or the decision to submit the work for publication.

## Author contributions

Kayson Fakhar, Conceptualization, Data curation, Formal analysis, Investigation, Visualization, Methodology, Writing – original draft, Writing – review and editing; Fatemeh Hadaeghi, Conceptualization, Formal analysis, Validation, Investigation, Methodology, Writing – original draft, Writing – review and editing; Caio Seguin, Conceptualization, Data curation, Investigation, Methodology, Writing – review and editing; Shrey Dixit, Investigation, Writing – review and editing; Arnaud Messé, Conceptualization, Formal analysis, Supervision, Methodology, Writing – review and editing; Gorka Zamora-López, Bratislav Misic, Conceptualization, Supervision, Writing – review and editing; Claus C Hilgetag, Conceptualization, Resources, Supervision, Funding acquisition, Validation, Writing – review and editing

## Author ORCIDs

Kayson Fakhar  https://orcid.org/0000-0003-0615-1777
Arnaud Messé  https://orcid.org/0000-0001-9081-3088
Bratislav Misic  https://orcid.org/0000-0003-0307-2862

Reviewer #2 (Public review): https://doi.org/10.7554/eLife.101780.3.sa1
Author response https://doi.org/10.7554/eLife.101780.3.sa2

---

# Additional files

## Supplementary files

MDAR checklist

## Data availability

The simulation results are available at the link below: https://zenodo.org/records/10849223. All connectomes are available as a part of the Netneurotools library: https://github.com/netneurolab/netneurotools (copy archived at *Bazinet et al., 2025*).

The following dataset was generated:

| Author(s) | Year | Dataset title | Dataset URL | Database and Identifier |
|---|---|---|---|---|
| Fakhar K | 2025 | A general framework for characterizing optimal communication in brain networks | https://doi.org/10.5281/zenodo.10849223 | Zenodo, 10.5281/zenodo.10849223 |

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
