## [Editor Report · eLife Assessment]

The authors provide a **compelling** method for characterizing communication within brain networks. The study engages **important**, biologically pertinent, concerns related to the balance of dynamics and structure in assessing the focal points of brain communication. It will be of interest to researchers trying to dissect structure of complex interaction networks across scales, from cells to regions.

---

## [Referee Report · Reviewer #2 (Public review)]

Summary:

The authors provide a compelling method for characterizing communication within brain networks. The study engages important, biologically pertinent, concerns related to the balance of dynamics and structure in assessing the focal points of brain communication. The methods are clear, and seem broadly applicable, although they require some forethought about data and modeling choices.

Strengths:

The study is well-developed, providing overall clear exposition of relevant methods, as well as in-depth validation of the key network structural and dynamical assumptions. The questions and concerns raised in reading the text were always answered in time, with straightforward figures and supplemental materials.

Weaknesses:

In earlier drafts of the work, the narrative structure at times conflicts with the interpretability, however, this was greatly improved during revisions. The only remaining limitation for broad applicability lies in the full observability required in the current paradigm, however, the authors point at avenues for relaxing this assumption, which could be fruitful next steps for researchers aiming to deploy this work to EM or two-photon based datasets.

---

## [Author Response]

The following is the authors’ response to the original reviews.

**Public Reviews:**

**Reviewer #1 (Public reviews):**
Summary:In this study, Fakhar et al. use a game-theoretical framework to model interregional communication in the brain. They perform virtual lesioning using MSA to obtain a representation of the influence each node exerts on every other node, and then compare the optimal influence profiles of nodes across different communication models. Their results indicate that cortical regions within the brain's "rich club" are most influential.Strengths:Overall, the manuscript is well-written. Illustrative examples help to give the reader intuition for the approach and its implementation in this context. The analyses appear to be rigorously performed and appropriate null models are included.

Thank you.

Weaknesses:The use of game theory to model brain dynamics relies on the assumption that brain regions are similar to agents optimizing their influence, and implies competition between regions. The model can be neatly formalized, but is there biological evidence that the brain optimizes signaling in this way? This could be explored further. Specifically, it would be beneficial if the authors could clarify what the agents (brain regions) are optimizing for at the level of neurobiology - is there evidence for a relationship between regional influence and metabolic demands? Identifying a neurobiological correlate at the same scale at which the authors are modeling neural dynamics would be most compelling.

This is a fundamental point, and we put together a new project to address it. The current work focuses on, firstly, rigorously formalizing a prevailing assumption that brain regions optimize communication, and then uncovering what are the characteristics of communication if this optimization is indeed taking place. Based on our findings, we suspect the mechanism of an optimal communication to be through broadcasting (compared to other modes explored in our work, e.g., the shortest-path signalling or diffusion). However, we recognize that our game-theoretical framework does not directly address “how” this mechanism is implemented. Thus, in our follow-up work, we are analyzing available datasets of signal propagation in the brain to see if communication dynamics there match the predictions of the game-theoretical setup. However, following your question, we extended our discussion to cover this point, cited five other works on this topic, and what, we think, could be the neurobiological mechanism of optimal signalling.

It is not entirely clear what Figure 6 is meant to contribute to the paper's main findings on communication. The transition to describing this Figure in line 317 is rather abrupt. The authors could more explicitly link these results to earlier analyses to make the rationale for this figure clearer. What motivated the authors' investigation into the persistence of the signal influence across steps?

Great question. Figure 6 in part follows Figure 5, which summarizes a key aspect of our work: Signals subside at every step but not exponentially (Figure 5), and they nearly fall apart after around 6 steps (Figure 6 A and B). Subplots A and B together suggest that although measures like communicability account for all possible pathways, the network uses a handful instead, presumably to balance signalling robustness versus the energetic cost of signalling. Subplot C, one of our main findings, then shows how one simple model is all needed to predict a large portion of optimal influence compared to other models and variables. In sum, Figure 5 focused on the decay dynamics while Figure 6 focused on the extent, in terms of steps, given that the decay is monotonic. Together, our motivation for this figure was to show how the right assumption about decay rate and dynamics can outperform other measures in predicting optimal communication.

The authors used resting-state fMRI data to generate functional connectivity matrices, which they used to inform their model of neural dynamics. If I understand correctly, their functional connectivity matrices represent correlations in neural activity across an entire fMRI scan computed for each individual and then averaged across individuals. This approach seems limited in its ability to capture neural dynamics across time. Modeling time series data or using a sliding window FC approach to capture changes across time might make more sense as a means of informing neural dynamics.

We agree with you on the fact that static fMRI is limited in capturing neural dynamics. However, we opted not to perform dynamic functional connectivity fitting just yet for a practical reason: Other communication models used here do not fit to any empirical data and provide a static view of the dynamics, comparable to the static functional connectivity. Since one of our goals was to compare different communication regimes, and the fact that fitting dynamics does not seem to substantially change the outcome if the end result is static (Figure 7), we decided to go with the poorer representation of neural data for this work. However, part of our follow-up project involves looking into the dynamics of influence over time and for that, we will fit our models to represent more realistic dynamics.

The authors evaluated their model using three different structural connectomes: one inferred from diffusion spectrum imaging in humans, one inferred from anterograde tract tracing in mice, and one inferred from retrograde tract-tracing in macaque. While the human connectome is presumably an undirected network, the mouse and macaque connectomes are directed. What bearing does experimentally inferred knowledge of directionality have on the derivation of optimal influence and its interpretation?

In terms of if directionality changes the interpretation of optimal influence, we think it sets limits for how much we can compare communication dynamics of these two types of networks. We think interpreting optimal communication in directed graphs needs to disentangle incoming influence from outgoing influence, e.g., analyzing “projector hubs/coordinators” and “receiver hubs/integrators” instead of putting both into a common class of hubs. Also, here we showed the extent of which a signal travels before it significantly degrades, having done so in an undirected graph. One of its implications for a directed graph is the possibility that some nodes can be unreachable from others, given the more restricted navigation. A possibility that we did not observe in the human connectome as all nodes could reach others, although with limited influence (see Figure 2. C). We did not explore these differences, as we used mice and macaque connectomes primarily to control for modality-specific confounds of DSI. However, our relatively poorer fit for directed networks (Supplementary Figure 2) motivated us to analyze how reciprocal connections shape dynamics and what impact do they have on networks’ function. Using the same connectomes as the current work, we addressed this question in a separate publication (Hadaeghi et al., 2024) and plan to extend both works by analyzing the signalling properties of directed networks.

It would be useful if the authors could assess the performance of the model for other datasets. Does the model reflect changes during task engagement or in disease states in which relative nodal influence would be expected to change? The model assumes optimality, but this assumption might be violated in disease states.

This is a wonderful idea that we initially had in mind for this work as well, but decided to dedicate a separate work on deviations in different tasks states, as well as disease states (mainly neurodegenerative disorders). We noticed the practical challenges of fitting large-scale models to task dynamics and harmonizing neuroimaging datasets of neurodegenerative disorders is beyond the scope of the current work. Unfortunately, this effort, although exciting and promising, is still pending as the corresponding author does not yet have the required expertise of neuroimaging processing pipelines.

The MSA approach is highly computationally intensive, which the authors touch on in the Discussion section. Would it be feasible to extend this approach to task or disease conditions, which might necessitate modeling multiple states or time points, or could adaptations be made that would make this possible?

Continuing our response from the previous point, yes, we think, in theory, the framework is applicable to both settings. Currently, our main point of concern is not the computational cost of the framework but the harmonization of the data, to ensure differences in results are not due to differences in preprocessing steps. However, assuming that all is taken care of, we believe a reasonable compute cluster should suffice by parallelizing the analytical pipeline over subjects. We acknowledge that the process would still be time-consuming, but besides the fitting process, we expect a modern high-performance CPU with about 32–64 threads to take up to 3 days analyzing one subject, given 100 brain regions or fewer. This performance then scales with the number of cluster nodes that can each work on one subject. We note that the analytical estimators such as SAR could be used instead, as it largely predicts the results from MSA. The limitations are then the lack of dynamics over time and potential estimation errors.

**Reviewer #2 (Public review):**
Summary:The authors provide a compelling method for characterizing communication within brain networks. The study engages important, biologically pertinent, concerns related to the balance of dynamics and structure in assessing the focal points of brain communication. The methods are clear and seem broadly applicable, however further clarity on this front is required.Strengths:The study is well-developed, providing an overall clear exposition of relevant methods, as well as in-depth validation of the key network structural and dynamical assumptions. The questions and concerns raised in reading the text were always answered in time, with straightforward figures and supplemental materials.

Thank you.

Weaknesses:The narrative structure of the work at times conflicts with the interpretability. Specifically, in the current draft, the model details are discussed and validated in succession, leading to confusion. Introducing a "base model" and "core datasets" needed for this type of analysis would greatly benefit the interpretability of the manuscript, as well as its impact.

Following your suggestion, we modified the introduction to emphasize on the human connectome and the linear model as the main toolkit. We also added a paragraph explaining the datasets that can be used instead.

**Recommendations for the authors:**

**Essential Revisions (for the authors):**
(1) The method presents an important and well-validated method for linking structural and functional networks, but it was not clear precisely what the necessary data inputs were and what assumptions about the data mattered. To improve the clarity of the presentation for the reader, it would be beneficial to have an early and explicit description of the flow of the method - what exact kinds of datasets are needed and what decisions need to be made to perform the analysis. In addition, there were questions about how the use or interpretation of the method might change with different methods of measuring structure or function, which could be answered via an explicit discussion of the issue. For example, how do undirected fMRI correlation networks compare to directed tracer injection projection networks? Similarly, could this approach apply in cases like EM connectomics with linked functional imaging that do not have full observability in both modalities?

This is an important point that we missed addressing in detail in the original manuscript. Now we did so, by first adding a paragraph (lines 292-305, page 10) explaining the pipeline and how our framework handles different modeling choices, and then further discussing it in the Discussion (lines 733-748, page 28). Moreover, we adjusted Figure 1, by delineating two main steps of the pipeline. Briefly, we clarified that MSA is model-agnostic, meaning that, in principle, any model of neural dynamics can be used with it, from the most abstract to the most biologically detailed. Moreover, the approach extends to networks built on EM connectomics, tract-tracing, DTI, and other measures of anatomical connectivity. However, we realized that a key detail was not explicitly discussed (pointed to by Reviewer #2), that is, the fact that these models naturally need to be fitted to the empirical dataset, even though this fitting step appears not to be critical, as shown in Figure 7.

Lines 292-305:

“The MSA begins by defining a ‘game.’ To derive OSP, this game is formulated as a model of dynamics, such as a network of interacting nodes. These can range from abstract epidemic and excitable models (Garcia et al., 2012; Messé et al., 2015a) to detailed spiking neural networks (Pronold et al., 2023) and to mean-field models of the whole brain dynamics, as chosen here (see below). The model should ideally be fitted to reflect real data dynamics, after which MSA systematically lesions all nodes to derive the OSP. Put together, the framework is general and model-agnostic in the sense that it accommodates a wide range of network models built on different empirical datasets, from human neuroimaging and electrophysiology to invertebrate calcium imaging, and anything in between. In essence, the framework is not bound to specific modelling paradigms, allowing direct comparison among different models (e.g., see section Global Network Topology is More Influential Than Local Node Dynamics).”

Lines 733-740:

“As noted in the introduction, OI is model-agnostic, here, we leveraged this liberty to compare signaling under different models of local dynamics, primarily built upon undirected human connectome data. We also considered different modalities, e.g., tract tracing in Macaque (see Structural and Functional Connectomes under Materials and Methods) to confirm that the influence of weak connections is not inflated due to imaging limitations (Supplementary Figure 5. A). The game theoretical formulation of signaling allows for systematic comparison among many combinations of modeling choices and data sources.”

We then continued with addressing the issue of full observability. We clarified that in this work, full observability was assumed. However, the mathematical foundations of our method capture unobserved contributors/influencers as an extra term, similar to the additive error term of a linear regression model. To keep the paper as non-technical as possible, we omitted expanding the axioms and the proof of how this is achieved, and instead referred to previous papers introducing the framework.

Lines 740-748:

“Nonetheless, in this work, we assumed full observability, i.e., complete empirical knowledge of brain structure and function that is not necessarily practically given. Although a detailed investigation of this issue is needed, mathematical principles behind the method suggest that the framework can isolate the unobserved influences. In these cases, activity of the target node is decomposed such that the influence from the observed sources is precisely mapped, while the unobserved influences form an extra term, capturing anything that is left unaccounted for, see (Algaba et al., 2019b; Fakhar et al., 2024) for more technical details.”

(2) The value of the normative game theoretic approach was clear, but the neurobiological interpretation was less so. To better interpret the model and understand its range of applicability, it would be useful to have a discussion of the potential neurobiological correlates that were at the same level of resolution as the modeling itself. Would such an optimization still make sense in disease states that might also be of interest?

This is a brilliant question, which we decided to explore further in separate studies. Specifically, the link between optimal communication and brain disorders is a natural next step that we are pursuing. Here, we expanded our discussion with a few lines first explaining the roots of our main assumption, which is that neurons optimize information flow, among other goals. We then hypothesized that the biological mechanisms by which this goal is achieved include (based on our findings) adopting a broadcasting regime of signaling. We suspect that this mode of communication, operationalized on complex network topologies, is a trade-off between robust signaling and energy efficiency. Currently, we are planning practical steps to test this hypothesis.

Lines 943-962:

“Nonetheless, our framework is grounded in game theory where its fundamental assumption is that nodes aim at maximizing their influence over each other, given the existing constraints. This assumption is well explored using various theoretical frameworks (Buehlmann and Deco, 2010; Bullmore and Sporns, 2012; Chklovskii et al., 2002; Laughlin and Sejnowski, 2003; O’Byrne and Jerbi, 2022) and remains open to further empirical investigation. Here, we used game theory to mathematically formalize a theoretical optimum for communication in brain networks. Our findings then provide a possible mechanism for achieving this optimality through broadcasting. Based on our results, we speculate that, there exists an optimal broadcasting strength that balances robustness of the signal with its metabolic cost. This hypothesis is reminiscent of the concept of brain criticality, which suggests the brain to be positioned in a state in which the information propagates maximally and efficiently (O’Byrne and Jerbi, 2022; Safavi et al., 2024). Together, we suggest broadcasting to be the possible mechanism with which communication is optimized in brain networks, however, further research directions include investigating whether signaling within brain networks indeed aligns with a game-theoretic definition of optimality. Additionally, if it does, subsequent studies could then examine how deviations from optimal communication contribute to or result from various brain states or neurological and psychiatric disorders.”

**Reviewer #1 (Recommendations for the authors):**
I would recommend that the authors consider the following point in a revision, as well as the major weaknesses of the public review. Some aspects of Figure 1 could be clearer. What is being illustrated by the looping arrow to MSA? What is being represented in the matrices (labeling "source" and "target" on the matrix might enhance clarity)? Is R2 the metric used to assess the degree of similarity between communication models? These could be addressed by making small additions to the figure legend or to the figure itself.

Thank you for your constructive comment on Figure 1, which is arguably the most important figure in the manuscript. We adjusted the figure and its caption (see above) based on your suggestions. After doing so, we think the figure is now clearer regarding the pipeline used in this work.

**Reviewer #2 (Recommendations for the authors):**
Overall, as stated in the public review and the short assessment, the manuscript is in a clearly mature state and brings an important method to link the fields of structural and functional brain networks.Nevertheless, the paper would benefit from an early, and clear, discussion of the:(1) components of the model, and assumptions of each, should be stated at the end of the introduction, or early in results. (2) datasets necessary to run the analysis.The confusion arises from lines 130-131, stating "In the present work (summarized in Figure 1), we used the human connectome, large-131 scale models of dynamics, and a game-theoretical perspective of signaling." This, to me, indicated that a structural connectivity map may be the only dataset required, as the dynamics model and game theory component are solely simulated. However, later, lines 214-216 state that the empirical functional connectivity is estimated from the structural connectivity, indicating that the method is only applied to cases where we have both.Finally, Supplemental Figure 5 validates a number of metrics on different solely structural networks (which is a very necessary and well-done control). Similarly, while the dynamical model is discussed in depth, and beautifully shown that the specific choice of dynamical model does not directly impact the results, it would be helpful to clarify the dynamical model utilized in the early figures.

Thank you for pointing out a critical detail that we missed elaborating sufficiently early in the paper: the modelling step. Following your suggestions, we added a paragraph from line 292 to 305 (page 10) expanding on the modelling framework. We also explicitly divided the modelling step in Figure 1 and briefly clarified our modelling choices in the caption. Together, we emphasized the fact that our framework is generally model agnostic, which allows different models of dynamics to be plugged into various anatomical networks. We then clarified that, like in any modelling effort, one needs to first fit/optimize the model parameters to reproduce empirical data. In other words, we emphasized the fact that our framework relies on a computational model as its ‘game’ to infer how regions interact, and we fine-tuned our models to reproduce the empirical FC.

Again, this is not a critique of the methods, which are excellent, but the presentation. It would help readers, and even me, to have a clear indication of the model earlier. Further, it would help to discuss, both in the introduction and discussion, the datasets required for applying these methods more broadly. For instance, 2-photon recordings are discussed - would it be possible to apply this method then to EM connectomes with functional data recorded for them? In theory, it seems like yes, although the current datasets have 100% observability, whereas 2-photon imaging, or other local methods, will not have perfect overlap between structural and functional connectomes. Discussions like this, related to the assumptions of the model, the necessary datasets, and broader application directions beyond DSI, fMRI, and BOLD cases where the method was validated, would increase the impact and interpretability for a broad readership.

This is a valid point that we should have been more explicit about. The revised manuscript now contains a paragraph (lines 740-748) clarifying the fact that, throughout this work, we assumed full observability. We then briefly discuss, based on the mathematical principles of the framework, what we expect to happen in cases with partial observability. We then point at two references in which the details of a framework with partial observability are laid out, one containing mathematical proofs and the other using numerical simulations.

References:

Hadaeghi, F., Fakhar, K., & Hilgetag, C. C. (2024). Controlling Reciprocity in Binary and Weighted Networks: A Novel Density-Conserving Approach (p. 2024.11.24.625064). bioRxiv. https://doi.org/10.1101/2024.11.24.625064